# Label Noise: Correcting the Forward-Correction

## Abstract

Training neural network classifiers on datasets with label noise poses a risk of overfitting them to the noisy labels. To address this issue, researchers have explored alternative loss functions that aim to be more robust. The 'forward-correction' is a popular approach wherein the model outputs are noised before being evaluated against noisy data. When the true noise model is known, applying the forward-correction guarantees the consistency of the learning algorithm. While providing some benefit, the correction is insufficient to prevent overfitting to finite noisy datasets. This work proposes an approach to tackling overfitting caused by label noise. We observe that the presence of label noise implies a lower bound on the noisy generalised risk. Motivated by this observation, we propose imposing a lower bound on the training loss to mitigate overfitting. Our main contribution is providing theoretical insights that allow us to approximate the lower bound, given only an estimate of the average noise rate. We empirically demonstrate that using this bound significantly enhances robustness in various settings with virtually no additional computational cost.

## 1 Introduction

Over the last decade, we have seen an enormous improvement in the efficacy of machine learning methods for classification. Correspondingly, there has been an increased need for large labelled datasets to train these models. However, obtaining cleanly labelled datasets at the required scale can be prohibitively costly. Consequently, practitioners often resort to methods that generate large but noisy datasets, such as web querying or crowd-sourcing. Label noise can be a particular issue when data are hard to label, or labelling requires a specialist background (e.g., medical imaging). This challenge has spurred significant interest in developing methods capable of learning with noisy labels.

Most approaches for addressing the label noise problem consist of a mechanism for either removing or compensating for it. However, many strategies to manage label noise are elaborate or require multiple networks and stages (Li et al., 2020; Han et al., 2018; Jiang et al., 2018; Malach and Shalev-Shwartz, 2017; Li et al., 2023; Ren et al., 2018; Sachdeva et al., 2021), which may not be feasible in resource-constrained settings.

A simpler approach designs methods to be inherently resilient in the face of corrupted labels. The most prominent family of such methods is *robust loss functions*. The goal is to choose an objective function which allows training in the presence of noise without harming the generality of the learned classifier. Regularisation-based robust loss functions use techniques like adding regularisation terms or consistency constraints to prevent overfitting (Zhang et al., 2017; Liu et al., 2020; Englesson and Azizpour, 2021b; Reed et al., 2014). Other regularisation-based losses adapt cross-entropy loss to reduce noise sensitivity (Wang et al., 2019; Zhang and Sabuncu, 2018; Ma et al., 2020). However, the robustness of such losses is often based on empirical observations rather than theoretical analysis, leaving the reasons for their effectiveness somewhat unclear.

*Loss correction methods* are a more principled subset of robust losses, using data to infer and subtract the impact of label noise from the training objective. Forward-corrections apply noise to the model predictions before evaluating these noised predictions against noisy data, whereas backward-corrections de-noise the noisy labels (Patrini et al., 2017). Despite satisfying slightly weaker theoretical properties than the backward-correction (Van Rooyen et al., 2015), forward-corrections generally perform slightly better, are easier to

implement, and are more frequently used (Patrini et al., 2017; Sukhbaatar et al., 2015; Goldberger and Ben-Reuven, 2022; Larsen et al., 1998; Mnih and Hinton, 2012; Li et al., 2021). Nevertheless forward-corrections are still susceptible to overfitting on small noisy datasets.

In this paper, we tackle the challenge of overfitting in popular robust losses, focusing on forward-corrected loss functions due to their prevalence and importance. We introduce a principled solution: bounding the allowable loss during training by recognising that the presence of label noise means the generalised noisy risk is lower bounded. The critical contribution of this paper is explicitly deriving these bounds and showing that their implementation improves robustness. In addition, we provide a deeper understanding of existing robust loss functions, unifying correction losses with several other popular regularisation-based losses into a single family.

**Key Idea:** When a distribution contains label noise, this implies there is a minimum achievable (noisy) risk. Current methods do not respect this bound, targeting a minimal training loss instead, causing overfitting. By bounding the loss below during training, one may prevent this.

## 1.1 Contributions

This work comprises two contributions: **(a)** a minor contribution is to **generalise the forward-correction** to include non-linear models, demonstrating that some popular regularisation-based robust loss functions (such as GCE and SCE) are, in fact, disguised forward-correction loss functions. This improves the generality of the main result where we: **(b)** show how overfitting can be avoided by ensuring the training loss remains above a certain threshold. We refer to augmenting a loss function in this manner as a **'bounded-loss'**. The crucial insight of this work is that when labels are noisy, no model can achieve a loss below the average entropy of the noisy conditional class distribution. Under a separability assumption, the noisy entropy depends only on the noise model and can thus be crudely estimated when only the average noise rate is known. This estimate, called the 'noise-bound', is chosen as our threshold for the bounded loss. We demonstrate empirically that this significantly enhances robustness across a broad range of settings.

$$\underbrace{\frac{1}{N}\sum_{i=1}^{N} L(\boldsymbol{q}(x_i), y_i)}_{\text{CE}} \xrightarrow{\text{forward-correct}} \underbrace{\frac{1}{N}\sum_{i=1}^{N} L(\widehat{T}\boldsymbol{q}(x_i), y_i)}_{\text{FCE}} \xrightarrow{\text{noise-bound}} \underbrace{\left\| B(\eta, c) - \frac{1}{N}\sum_{i=1}^{N} L(\widehat{T}\boldsymbol{q}(x_i), y_i) \right\|_1}_{\text{FCE+B}} \quad (1)$$

Figure 1: Cross-entropy (CE), forward-corrected cross-entropy (FCE) and, our proposed FCE+B loss functions. The forward-correction ensures consistency, while the application of a bound (FCE+B) mitigates overfitting.

## 1.2 Notation and Preliminaries

We adhere to standard notational and terminological conventions used in related literature. Key notations are defined below; a detailed summary and review of **related work** are available in Appendix B.

**Domains** $\mathcal{X} \subset \mathbb{R}^d$ denotes the dataspace, and $\mathcal{Y} := \{1, 2, \ldots, c\}$ represents the label space for $c$ classes. The probability simplex, $\Delta$, consists of $c$-dimensional vectors with non-negative components that sum to one. Vectors are indicated in **bold**.

**Loss Function** A loss function $L : \Delta \times \mathcal{Y} \to \mathbb{R}$ measures the discrepancy between predicted and actual label distributions, resulting in a loss value. The loss across all classes for a prediction $\boldsymbol{q}$ is expressed as:

$$\boldsymbol{L}(\boldsymbol{q}) = (L(\boldsymbol{q}, 1), L(\boldsymbol{q}, 2), \ldots, L(\boldsymbol{q}, c)).$$

**Expected Loss**   The *expected loss* for a forecast $\boldsymbol{q}$ relative to the true class distribution $\boldsymbol{p}$ is the average loss over labels sampled from $\boldsymbol{p}$, denoted by $H_L(\boldsymbol{p}, \boldsymbol{q})$ and calculated via:

$$H_L(\boldsymbol{p}, \boldsymbol{q}) = \boldsymbol{p}^T \boldsymbol{L}(\boldsymbol{q}).$$

**Proper Losses**   A loss $L$ is considered (strictly) proper if the expected loss is (uniquely) minimised by the true label distribution, i.e., $\boldsymbol{q} := \boldsymbol{p}$.

**$L$-Risk**   The (generalised) $L$-risk, representing the expected loss over the *entire* data distribution $p(x, y)$, evaluates the overall efficacy of a probability estimator model $\boldsymbol{q}(x)$:

$$R_L(\boldsymbol{q}) := \mathbb{E}_{x \sim p(x)}[H_L(\boldsymbol{p}(y \mid x), \boldsymbol{q}(x))].$$

 We call the risk computed with respect to an i.i.d dataset drawn from $p(x, y)$, the *empirical* risk, which we denote $\widehat{R}_L$.

**Label Noise**   Label noise refers to any random process altering labels from the data-label distribution $p(x, y)$. We use a tilde notation to distinguish between original (clean) and altered (noisy) labels, such as $\widetilde{p}(x, \widetilde{y})$. For a detailed taxonomy of label noise types, see Appendix A.

**Noisy Risk**   Label noise introduces the concept of *noisy* generalised risk, denoted by $R_L^{\eta}$. This risk is computed with respect to the noisy label distribution.

**Noise Rate**   The *noise rate* at any $x \in \mathcal{X}$, defined as $\eta(x) := p(\widetilde{y} \neq y \mid x)$, reflects the likelihood of label alteration by noise at that point. The *average noise rate* across the dataspace is $\eta := p(\widetilde{y} \neq y)$.

## 2   Generalised Forward-Corrections

### 2.1   Robust Loss Functions

Label noise robust loss functions can be categorised into two broad sets: regularisation-based robust losses and correction-based losses.

**Regularisation-Based Losses**   A popular approach to tackling label noise by selecting losses less prone to fit the entire training set than the standard cross-entropy (CE). An archetypal example of such a loss is a mean absolute error (MAE) ($L_{MAE}(\boldsymbol{q}, y = k) = 1 - q_k$). MAE will typically ignore the harder-to-fit samples; on noisy datasets, this often corresponds to those with corrupted labels. The downside is that MAE dramatically underfits on datasets with many classes (Ma et al., 2020). Alternative losses mitigate this underfitting by interpolating between CE and MAE to avoid both of their pitfalls. Two well-known examples are the Generalised Cross-Entropy (GCE) and Symmetric Cross-Entropy (SCE) defined $L_{GCE}(\boldsymbol{q}, y = k) := \frac{1 - q_k^a}{a}$ and $L_{SCE}(\boldsymbol{q}, y = k) = -log(q_k) + A(1 - q_k)$ respectively (Zhang and Sabuncu, 2018; Wang et al., 2019). By varying the parameters $a, A$, we can alter the losses' behaviour from being more like CE to MAE.

**Correction-based Losses**   Correction-based loss functions arise as an alternative, motivated by the observation that the empirical risk ceases to be an effective proxy for the generalised clean risk (Stempfel and Ralaivola, 2009) under label noise. By altering the loss through incorporating the noise model, one may ensure the corrective property that

$$\arg\min_{\boldsymbol{q}} R_{L_F}^{\eta}(\boldsymbol{q}) = \arg\min_{\boldsymbol{q}} R_L(\boldsymbol{q}),$$

ensuring that minimising the noisy generalised risk aligns with minimising the generalised clean risk. A popular and effective method is the *forward-correction* (Patrini et al., 2017). Given a *base loss* $L$, the forward-correction of $L$ is defined

$$L_F(\boldsymbol{q}, k) := L(\widehat{T}\boldsymbol{q}, k), \tag{2}$$

where $\widehat{T}$ is a column stochastic matrix approximating the true transition matrix $T := p(\widetilde{y}|y)$ for class-conditional label noise. Conversely, we say that a loss function $L$ is a *forward-corrected loss* if it is the forward-correction of another loss function.

## 2.2 Non-Linear Noise Models

In the class-conditional label noise framework (Angluin and Laird, 1988), a label noise model is defined by a column stochastic matrix $T$, which represents the transition probabilities ($T_{ij} \coloneqq p(\widetilde{y} = j \mid y = i)$). This traditional formulation assumes that the noisy label depends on an unobserved clean label, which might not reflect real-world scenarios where the noisy annotator never sees the clean label.

An equivalent conceptualisation of class-conditional label noise considers the transition matrix merely as a tool to link noisy and clean class distributions. For example, if the true conditional class distribution at $x$ is $\boldsymbol{p}(y \mid x)$, then according to a class-conditional label noise model, the conditional class distribution of the noisy labeller is given by $T\boldsymbol{p}(y \mid x)$. This approach does not presume dependence on a specific true label, instead describing how noisy and clean label distributions are related.

However, this perspective raises questions about the necessity of assuming a linear relationship between these distributions. It is plausible that a labeller might make fewer errors on instances clearly representative of their class and many more errors when the class distribution $\boldsymbol{p}(y \mid x)$ is more evenly distributed. This scenario suggests a non-linear noise model:

$$\widetilde{\boldsymbol{p}}(y \mid x) \coloneqq f(\boldsymbol{p}(y \mid x)),$$

where $f : \Delta \to \Delta$ is some (possibly non-linear) transformation on this simplex - which, for simplicity, we will limit to being injective.

It is straightforward to generalise the forward-correction to allow for $\widehat{T}$ in Equation 2 being a non-linear transformation $f$.

**Definition 2.1** (Generalised Forward-Correction). Let $L_f$ be a loss function and $f : \Delta \to \Delta$ be an injective function. We say $L_f$ is a 'generalised forward-corrected loss' if there exists a loss function $L$ such that for all $\boldsymbol{q} \in \Delta$, $k \in \{1, 2, \ldots, c\}$

$$L_f(\boldsymbol{q}, k) = L(f(\boldsymbol{q}), k).$$

We refer to $L$ as the **base loss**. $f$ can be thought of as a label noise model.

The forward-correction is trivially an example of a generalised forward-correction loss obtained by setting $f \coloneqq \widehat{T}$. We now demonstrate that the GCE and SCE losses previously discussed are generalised forward-correction losses, deriving expressions for the underlying (non-linear) noise models $f$. This derivation relies on the assumption that the base losses which generate these loss functions are proper - i.e. we decompose both SCE and GCE into a proper loss corrected by a non-linear noise model. This decomposition is unique.

**Lemma 2.2.** The GCE, SCE and forward-corrected CE (denoted FCE) loss functions can be formulated as generalised forward-corrected losses with a proper base loss. The noise models $f_{GCE}, f_{SCE}, f_{FCE}$ satisfy

$$(f_{GCE}^{-1}(\boldsymbol{p}))_i = \frac{p_i^{\frac{1}{1-a}}}{\sum_{i=1}^{c} p_i^{\frac{1}{1-a}}}, \quad (f_{SCE}^{-1}(\boldsymbol{p}))_i = \frac{p_i}{\lambda - A p_i}, \quad f_{FCE}(\boldsymbol{p}) = \widehat{T}\boldsymbol{p},$$

where $\widehat{T}$ is the invertible stochastic matrix used to define FCE, and $\lambda$ is a constant selected to ensure the correct normalisation.

For an interpretation and plots of the noise models; $f_{GCE}, f_{SCE}$, see Appendix D.

Lemma 2.2 demonstrates that GCE and SCE can be conceptualised as non-linear forward-corrected losses; the noise model is represented by the function $f$. We stress that these are by no means the only robust losses which adhere to Definition 2.1. However, they provide valuable examples when empirically demonstrating the results of Section 4.

The generalisation established by Definition 2.1 offers three advantages. i) It enhances our understanding of losses like GCE, demonstrating that they implicitly incorporate a noise model. ii) Partially unifies correction losses with other robust loss functions. iii) Ensures that theoretical results derived for generalised forward-correction losses are widely applicable, encompassing traditional, linear correction losses and many other robust loss functions.

# 3 Loss Bounding

Our analysis in this section reveals that merely correcting for the noise model (as in Definition 2.1) is inadequate for achieving robustness. We must also adjust our loss function to incorporate a lower bound to account for the randomness introduced by label noise.

**Key Observation**  When a data distribution contains label noise, there is a lower bound on the optimal noisy risk a model can achieve. An analogy to this is that no forecaster can predict the outcome of a biased coin flip 100% of the time - e.g. if a coin comes up heads 60% of the time we cannot expect a forecaster to predict more accurately than 60% over a large number of flips. Similarly, even an optimal model, which minimises the noisy risk, will *still* incur a non-zero loss on a randomly sampled noisy dataset.

## 3.1 Overfitting to Label Noise

If one trains a classifier on a noisy dataset using the cross-entropy loss function, the classifier learns to fit all labels in the training set - including those which have been corrupted by label noise (Arpit et al., 2017) - damaging generality. Ideally, when training a model on a noisy dataset, we wish to fit the clean labels without overfitting the noisy ones. A significant obstacle to achieving this desire is the difficulty in determining whether a specific label is clean or corrupted. However, while it is difficult to determine whether a model has overfit to a *specific* label, it is often apparent when a model has overfit to a dataset. For example, if a dataset is known to contain label noise, obtaining a training loss of zero heavily implies overfitting has occurred.

**Forward-Corrections**  When learning a classifier in the presence of noisy labels, it is, therefore, inappropriate to target a zero training loss. Utilising the forward-correction partly addresses this issue. The forward-correction works by noising our model predictions $q \mapsto \widehat{T}q$ before applying the loss. This guarantees that a zero loss is no longer possible since our noised model never predicts any label with complete confidence. Despite possessing this desirable property, the forward-correction does not go far enough in that the lower bound it imposes is still too low to prevent overfitting. An illustrative example for a dataset polluted by 40% symmetric label noise is presented in Table 1. The table gives the lowest attainable training loss for a model trained on this noisy dataset versus the 'optimal training loss', i.e. the loss obtained by an optimal model possessing complete knowledge of how the dataset was generated but has not observed the dataset labels. Obtaining a training loss lower than this optimal value would suggest that overfitting has occurred. We see that while FCE imposes a bound (unlike CE), this bound is still too low. Further discussion of this example is given in Appendix D.1.

| Loss Function | Lowest Attainable Training Loss | Optimal Training Loss |
|:---:|:---:|:---:|
| CE | 0 | 0.673 |
| FCE | 0.511 | 0.673 |
| FCE+B | 0.673 | 0.673 |

Table 1: Lowest attainable training loss for different cross-entropy variants versus *optimal* training loss. This table contrasts the minimum training loss achievable by a model using various loss functions on a noisy dataset with 40% symmetric label noise. An optimal model minimises the noisy generalised risk, achieving a loss of 0.673. CE or FCE can result in lower training losses, potentially causing overfitting. While FCE introduces a loss bound to improve robustness, FCE+B is designed to mitigate overfitting more effectively.

## 3.2 Bounded Loss

We propose, therefore, that the principled way to handle label noise is to limit the minimum allowable risk on the training set. Specifically, we define a lower bound '$B$' and train - preventing the training loss from going below this value. Explicitly, we augment our loss as follows:

**Definition 3.1** (Bounded Loss)**.** Let $L$ be a loss function. Let $\mathcal{D}$ be a batch of $N$ data-label pairs $(x_i, y_i)$. Given a lower bound, $B \in \mathbb{R}$, we define the $B$-**bounded** loss $L_{\underline{B}}$ obtained from $L$ as follows:

$$L_{\underline{B}}(\boldsymbol{q}(x), \mathcal{D}) := \left|\left| B - \frac{1}{N}\sum_{i=1}^N L(\boldsymbol{q}(x_i), y_i) \right|\right|_1 \tag{3}$$

When the average loss on a batch of samples is above our bound $B$, training proceeds as usual; however, if the training loss on a batch dips below $B$, the learning rate effectively becomes negative, resulting in 'untraining' which proceeds until the average loss is back above $B$. Bounding the loss in this way has been previously explored by Ishida et al. (2020). We extend upon this work by grounding loss bounding within the context of loss corrections and by providing a theoretically justified method for selecting the loss bound.

## 4 Risk Bounds

In the last section, we noted that correction losses remain prone to overfitting despite their theoretical basis. We suggested training with a lower bound, acknowledging that the minimal achievable generalised noisy risk is non-zero. This section derives explicit lower bounds on the generalised noisy $L$-risk for forward-corrected losses and provides a formula for selecting a bound $B$, termed the *noise-bound*, for use in Definition 3.1.

**Assumptions** Throughout this section, we assume all loss functions are generalised forward-corrected, denoted $L_f$, with proper base losses $L$. We also assume that the loss function has no inherent bias toward any particular class, i.e. the loss is unaffected by a random permutation of the label set. Examples include CE, MSE, FCE, GCE, SCE, among others.

### 4.1 Entropy As Lower Bound

In Lemma 4.2 we establish a general lower bound on noisy risk in terms of the average entropy of the noisy label distribution. Precisely stating Lemma 4.2 requires us to define the 'entropy function' of a proper loss.

**Definition 4.1** (Entropy Function)**.** Given a proper loss function $L$, define its entropy function (Ovcharov, 2018) as the expected loss incurred when the forecast equals the true distribution over classes: $\mathcal{H} : \Delta \to \mathbb{R}$ by:

$$\mathcal{H}(\boldsymbol{p}) := H_L(\boldsymbol{p}, \boldsymbol{p}) = \boldsymbol{p}^T \boldsymbol{L}(\boldsymbol{p}),$$

where $\boldsymbol{p}$ is a probability distribution over the classes and $H_L$ denotes the expected loss.

The entropy function for a (strictly) proper loss function is (strictly) concave and, by the definition of properness, satisfies $\mathcal{H}(\boldsymbol{p}) \leq H(\boldsymbol{p}, \boldsymbol{q})$ for all $\boldsymbol{p}, \boldsymbol{q} \in \Delta$. This leads immediately to the following Lemma.

**Lemma 4.2.** Let $L_f$ be a generalised forward-corrected loss whose 'base-loss' $L$ is strictly proper (Recall the definition of 'base-loss' from Definition 2.1). The noisy risk of any probability estimator $\boldsymbol{q}$ is lower bounded:

$$R_{L_f}^{\eta}(\boldsymbol{q}) \geq \mathbb{E}_{x \sim p(x)}[\mathcal{H}(\widetilde{\boldsymbol{p}}(\tilde{y}|x))], \tag{4}$$

where $\mathcal{H}$ is the entropy function of the base-loss. This bound is tight when $f$ equals the true noise model. Equality is attained by setting $\boldsymbol{q}(x) = f^{-1}(\widetilde{\boldsymbol{p}}(\widetilde{y}|x))$.[1]

Lemma 4.2 establishes that when using a generalised forward-corrected loss (with proper base loss), the average entropy of the noisy distribution provides a lower bound on the noisy risk. Ideally, one would calculate the entropy in Equation 4 to use as the lower bound '$B$' in Equation 3. However, due to limited knowledge of the data distribution and noise model, it is often impractical to calculate this precisely, necessitating simplifying assumptions for approximation.

**Separability** Our key simplifying assumption is that the clean label distribution is (approximately) separable, meaning there's minimal randomness in label distributions. Although idealised, this assumption fits many real-world image classification tasks dominated by clear, single-subject images. We use this assumption to establish several bounds in the subsequent sections.

---

[1]Note that if $f$ is the true noise model then $\widetilde{\boldsymbol{p}}(\widetilde{y} \mid x) \in f(\Delta)$ and the inverse is unique by injectivity.

## 4.2 Estimating The Entropy

Given a datapoint $x$, the entropy of the noisy class distribution $\widetilde{\boldsymbol{p}}(\widetilde{y} \mid x)$ will depend both of the noise rate and the *type* of label noise. For example, symmetric label noise (where noise occurs uniformly between classes) will result in a different entropy than pairwise label noise (where noise occurs between pairs of classes), even given the same noise rate. The following Lemma gives a range on the possible entropies of $\widetilde{\boldsymbol{p}}(\widetilde{y} \mid x)$ given that the noise rate at $x$ is equal to $\eta(x)$. For brevity we introduce the following notation:

$$\boldsymbol{u}_{\mathrm{pair}}(\eta, c) := (1 - \eta, \eta, 0, \dots, 0) \tag{5}$$

$$\boldsymbol{u}_{\mathrm{sym}}(\eta, c) := \left(1 - \eta, \frac{\eta}{c-1}, \frac{\eta}{c-1}, \dots, \frac{\eta}{c-1}\right) \tag{6}$$

**Lemma 4.3.** Let $L_f$ be a generalised forward-corrected loss function whose base-loss $L$ has entropy function $\mathcal{H}$. Suppose that label noise is applied to a separable data-label distribution and let $x \sim p(x)$. Given that the noise rate at $x$ is $\eta(x)$, the entropy of the noisy label distribution at $x$, $\mathcal{H}(\widetilde{\boldsymbol{p}}(\widetilde{y} \mid x))$ must lie in the following interval:

$$\left[\mathcal{H}(\boldsymbol{u}_{\mathrm{pair}}(\eta(x), c)), \mathcal{H}(\boldsymbol{u}_{\mathrm{sym}}(\eta(x), c))\right].$$

In particular, given a specified noise rate $\eta(x)$, the highest entropy occurs under symmetric label noise at $x$, while the lowest entropy is observed with pairwise label noise.

**Corollary 4.4.** Given an average noise rate $\eta := \mathbb{E}_{x \sim p(x)}[\eta(x)]$, the greatest possible value of $\mathbb{E}_{x \sim p(x)}[\mathcal{H}(\widetilde{\boldsymbol{p}}(\widetilde{y} \mid x)]$ occurs when $\eta(x)$ is constant:

$$\sup_{p(\widetilde{y}|x,y)} \left(\mathbb{E}_{x \sim p(x)}[\mathcal{H}(\widetilde{\boldsymbol{p}}(\widetilde{y} \mid x))]\right) = \mathcal{H}\left(\boldsymbol{u}_{\mathrm{sym}}(\eta, c)\right),$$

where the supremum is taken over all noise models such that $\mathbb{E}_{x \sim p(x)}[\eta(x)] = \eta$.

**Worst-Case Entropy** Corollary 4.4 establishes a worst-cast entropy given a specified average noise rate $\eta$. Simply put, the Corollary tells us *'given that the average noise rate does not exceed $\eta$, the noise model with the highest entropy is uniform symmetric label noise'.*

## 4.3 Main Proposal: Noise-Bounded Loss

**Discussion** The goal of this section is to derive lower bounds on the noisy risk which can be used as '$B$' in our $B$-bounded loss (Definition 3.1). Ideally, with precise knowledge of the noise model, the lower bound would be set equal to the average entropy of the noisy label distribution. However, the label noise model is typically unknown, and we might only have access to an approximate noise rate.

**What bound do we choose when we only know the noise rate?** Corollary 4.4 establishes that, given a known noise rate $\eta$, a 'worst-case' entropy occurs when the label noise is symmetric and uniform. This means that if we set our bound '$B$' under the assumption of symmetric-uniform label noise at rate $\eta$, $B$ can never be lower than the actual noisy entropy - making overfitting unlikely. We call this the 'noise-bound'.

**Definition 4.5** (Noise-Bound). Let $L_f$ be a generalised forward-corrected loss whose base loss $L$ has entropy function $\mathcal{H}$. Using the notation $\boldsymbol{u}_{\mathrm{sym}}(\eta, c)$ from Equation 6, we define the **noise-bound** as:

$$\boxed{B(\eta, c) := \mathcal{H}(\boldsymbol{u}_{\mathrm{sym}}(\eta, c)) = \boldsymbol{u}_{\mathrm{sym}}(\eta, c) \cdot \boldsymbol{L}(\boldsymbol{u}_{\mathrm{sym}}(\eta, c)).} \tag{7}$$

**Examples** For CE/FCE the noise-bound corresponds to the Shannon Entropy of the distribution $\boldsymbol{u}_{\mathrm{sym}}(\eta, c) = (1 - \eta, \frac{\eta}{c-1}, \dots, \frac{\eta}{c-1})$. For SCE and GCE we remark that

$$B(\eta, c) = \boldsymbol{u}_{\mathrm{sym}}(\eta, c) \cdot \boldsymbol{L}(\boldsymbol{u}_{\mathrm{sym}}(\eta, c))$$
$$= \boldsymbol{u}_{\mathrm{sym}}(\eta, c) \cdot \boldsymbol{L}_f(f^{-1}(\boldsymbol{u}_{\mathrm{sym}}(\eta, c))),$$

enabling computation of $B(\eta, c)$ using expressions for $f^{-1}$ from Lemma 2.2, detailed in Appendix D.4.

This leads us to the main proposal of this paper. When our dataset has label noise, we propose using the bounded loss (Equation 3) with $B$ set to $B(\eta, c)$, the 'noise-bound' from Equation 7. We call this the **noise-bounded loss**:

**Definition 4.6** (Noise-Bounded Loss). Let $L$ be a loss function. Let $\mathcal{D}$ be a batch of $N$ data-label pairs $(x_i, y_i)$. Given a noise rate $\eta$, we define the **noise-bounded** loss $L_{B(\eta,c)}$ obtained from $L$ as follows:

$$L_{B(\eta,c)}(\boldsymbol{q}(x), \mathcal{D}) \coloneqq \left|\left| B(\eta,c) - \tfrac{1}{N} \sum_{i=1}^{N} L(\boldsymbol{q}(x_i), y_i) \right|\right|_1 \tag{8}$$

where $B(\eta, c)$ is as given in Equation 7. For an algorithmic implementation of this approach, see Algorithm 1.

**Example FCE:** The noise-bounded variant of FCE (which we denote FCE+B) is given in Equation 1.

**Bound Optimality** By construction, the noise-bound only equals the average entropy of the noisy label distribution if the noise is symmetric and uniform. In all other cases the noise-bound will be higher than strictly necessary. Ideally, we would like the gap between the noise-bound and the true noisy entropy to be small in a typical setting: If the gap is large, the noise-bounded loss will cease training long before overfitting occurs and possibly when there is still signal to be learned. A small gap occurs when all distributions of the form $(1 - \eta, \eta_2, \ldots, \eta_c)$ (where $\eta \coloneqq \sum_i \eta_i$) have roughly the same entropy - i.e. the entropy is 'insensitive' to the *structure* of the noise model, depending mostly on the noise *rate*. The level of insensitivity depends on the entropy function itself. Shannon entropy is relatively sensitive to the structure of the noise model. In contrast, losses like GCE and SCE induce insensitive entropy functions. Further discussion in Appendix D.2.

**Empirical and Generalised Risk** Our analysis in Section 4.1 demonstrates that an estimator cannot achieve a noisy risk below the mean noisy label entropy. However, on a small finite dataset of i.i.d. samples, it is possible for an estimator to achieve a noisy *empirical* risk that is lower than the noisy entropy, and this can occur with non-zero probability, even without access to the actual dataset labels. However, as the size of the dataset increases, the likelihood of an estimator achieving a loss significantly lower than the noisy entropy rapidly diminishes. According to the Central Limit Theorem, the probability of obtaining a loss more than $\delta$ below the noisy label entropy diminishes at the rate of $O\left(\frac{1}{\sqrt{N}}\right)$, where $N$ is the dataset size. In practical terms, this means obtaining a training loss below the noise-bound is *almost impossible* unless one has overfit to the noisy labels.

## 5 Experiments

### 5.1 Loss Functions

In this section, we empirically investigate the effectiveness of the noise-bounded loss (Equation 8) for improving robustness to label noise. We consider several loss functions: CE, SCE, forward-corrected CE (FCE), and GCE. Additionally, we explore a variant of CE that includes a prior on the model probabilities (CEP). Our experiments all follow a similar structure. We use a dataset containing intrinsic or synthetic label noise in the training set. We train neural network models using each loss on this noisy training set and evaluate their performance on a clean test set. We compare results from models trained without noise-bounds to those trained with noise-bounds, denoted by a '+B' suffix in the loss name (e.g., CE+B indicates the use of a noise-bounded cross-entropy loss).

**Baseline Loss Functions** Our results are benchmarked against other standard robust loss functions, including mean squared error (MSE) (Janocha and Czarnecki, 2017), mean absolute error (MAE), NCE-MAE (Ma et al., 2020), ELR (Liu et al., 2020), Curriculum loss (CL) (Zhou et al., 2020), Bootstrapping loss (Boot.) (Reed et al., 2014), Spherical loss (Spher.), Mix-up (Zhang et al., 2017), and a version of GCE that incorporates the additional tricks outlined by Zhang and Sabuncu (2018). To differentiate this version of GCE from our simplified GCE, we refer to it as 'Truncated loss' (Trunc.) due to its use of truncation.

## 5.2 Datasets

We evaluate each loss on various datasets with different label noise types. We consider versions of EMNIST, FashionMNIST, CIFAR10, CIFAR100 corrupted by symmetric label noise at rates of 0.2 and 0.4 and MNIST with rates of 0.4 and 0.6. Additionally, we explore more sophisticated noise types. In the case of 'Asym-CIFAR100,' we introduce asymmetric noise by randomly transitioning labels within the 20 superclasses of CIFAR100. For example, within the superclass 'fish' (comprised of aquarium-fish, flatfish, ray, shark, and trout), we change training labels to other members of the set with a probability of $\eta \in \{0.2, 0.4\}$ (e.g., flatfish $\rightarrow$ trout). For 'Non-Uniform EMNIST,' we investigate the impact of using non-uniform noise. We train a linear classifier on EMNIST and, with a probability of 0.6, modify the label of a data point in our training set to match the output of this classifier. Since the classifier's performance varies across data space, this creates label noise with an $x$-dependence. Further experiments on the TinyImageNet and Animals-10N datasets, which contain real, intrinsic open-set noise, are given in Appendix E.

**Hyperparameters** For the Animals and TinyImageNet experiments, we use a ResNet-34 to parameterise our model. For the other datasets, we use a ResNet-18. For each experiment, the number of epochs is kept consistent across losses. The bounds we employ in each experiment are obtained by substituting the relevant number of classes $c$ and the noise rate $\eta$ into Equation 7. An exception is the case of Non-uniform EMNIST, where we use a class number of $c = 2$ to reflect that the label is a mixture of the clean label and classifier labels. The results where we vary the bound '$B$' are obtained by doing a small grid search on either side of our noise-bound value. Additional precise experimental details may be found in Appendix E.

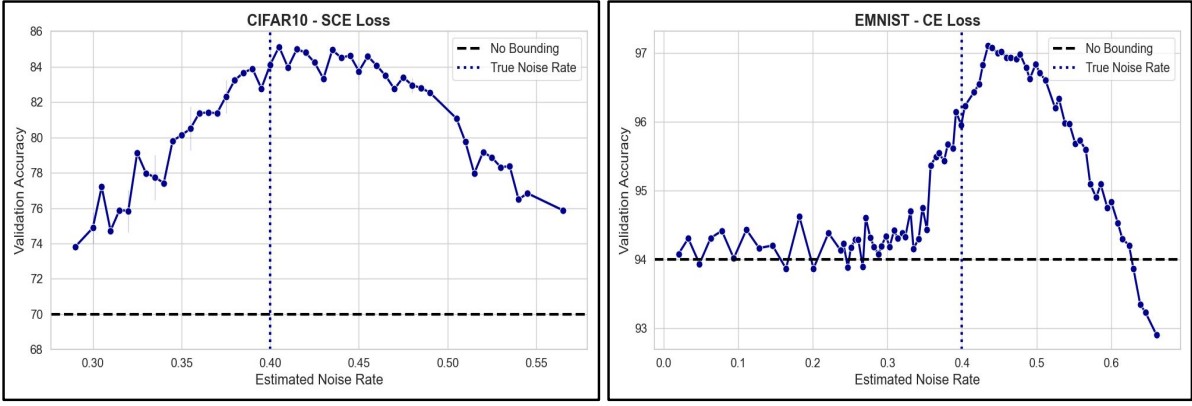

Figure 2: **Performance as a function of the estimated noise rate used to compute noise-bound:** We plot the final (clean) validation accuracy of a model against the estimated noise rate used to compute the noise-bound (Eqn. 7) on the noisy CIFAR10/EMNIST datasets using SCE/CE losses respectively. **The noise-bound, as computed with the *true* noise rate is highlighted by the green dotted line**; both graphs show a bump with a peak near this line demonstrating that underestimating the noise rate causes overfitting while overestimating causes underfitting. Most crucially, the prominent 'bump' reinforces that robustness can be greatly improved by training using a well-selected bound.

## 5.3 Results

The results of our experiments are presented in two tables. Table 2 includes the simpler datasets of MNIST, FashionMNIST, EMNIST, and CIFAR10, while Table 3 displays CIFAR100, Asym-CIFAR100, and Non-uniform-EMNIST. The results for TinyImageNet and Animals are presented in a third table in Appendix E. Each table follows a similar structure, with losses listed in rows and datasets in columns. The baselines are grouped together at the top. Our main losses are organised into pairs, such as CE, CE+B. The rows that use the noise-bound (e.g., GCE+B) are highlighted in blue to enhance readability. If using our noise-bound leads to higher mean accuracy compared to training without the bound, this is indicated by a $\boxed{box}$. The best

overall model for each dataset is highlighted in **bold**. In **82**% of cases utilising the noise-bound improves performance relative to the unbounded loss variant.

**Exceptions**  With few exceptions, our bound leads to improved performance compared to the standard, unbounded version of each loss. For the Asym-CIFAR100 and Non-Uniform-EMNIST datasets, our CE+B loss performs worse than regular CE. This outcome was expected since our derived bounds are optimal for symmetric noise and may be suboptimal for non-symmetric noise - this discrepancy is especially pronounced for losses based on Shannon-Entropy like CE. In contrast, the other generalised forward-corrected losses, as we had anticipated, exhibit greater resilience to the precise noise structure and consistently outperform the baseline across different types of noise.

**Impact of Estimated Noise Rate**  Figure 2 shows how clean test accuracy varies with the estimated noise rate, $\widehat{\eta}$, for CIFAR10 (SCE loss) and EMNIST (CE loss) datasets, both corrupted by 40% symmetric noise. Models were trained using noise-bounds based on $\widehat{\eta}$ ($B(\widehat{\eta}, c = 10)$ in Equation 7), and performance was plotted against $\widehat{\eta}$. As $\widehat{\eta}$ increases, the bound restricts overfitting, enhancing test accuracy. Optimal performance occurs near the true noise rate at $\widehat{\eta} \approx 0.4$, marked by a vertical green dotted line. Beyond this point, performance declines as the model underfits. The prominent peak in performance near this green line empirically validates our theoretical approach. Slight overestimations of the noise rate marginally improve performance, likely originating from our simplifying assumption, which modelled the underlying distributions as separable. Appendix E discusses this further, treating '$B$' as a hyperparameter.

| Losses | MNIST | | FashionMNIST | | EMNIST | | | | CIFAR10 | |
|---|---|---|---|---|---|---|---|---|---|---|
| | 0.4 | 0.6 | 0.2 | 0.4 | 0.2 Top 1 | Top 5 | 0.4 Top 1 | Top 5 | 0.2 | 0.4 |
| MSE | $93.3_{\pm 0.47}$ | $85.8_{\pm 0.95}$ | $84.8_{\pm 0.22}$ | $80.6_{\pm 0.84}$ | $82.9_{\pm 0.29}$ | $98.1_{\pm 0.04}$ | $80.2_{\pm 0.19}$ | $97.1_{\pm 0.07}$ | $78.7_{\pm 1.51}$ | $56.4_{\pm 0.11}$ |
| MAE | $97.9_{\pm 0.08}$ | $96.4_{\pm 0.08}$ | $83.2_{\pm 0.10}$ | $82.2_{\pm 0.37}$ | $49.8_{\pm 2.83}$ | $52.2_{\pm 0.10}$ | $50.4_{\pm 1.14}$ | $51.4_{\pm 0.96}$ | $88.6_{\pm 1.34}$ | $78.9_{\pm 5.95}$ |
| NCE | $97.8_{\pm 0.06}$ | $96.0_{\pm 0.25}$ | $87.7_{\pm 0.26}$ | $\mathbf{86.3}_{\pm 0.14}$ | $84.5_{\pm 0.25}$ | $97.9_{\pm 0.05}$ | $82.6_{\pm 0.81}$ | $96.7_{\pm 0.03}$ | $\mathbf{89.3}_{\pm 0.40}$ | $\mathbf{86.0}_{\pm 0.81}$ |
| MixUp | $95.8_{\pm 1.24}$ | $86.8_{\pm 0.85}$ | $86.9_{\pm 0.10}$ | $82.3_{\pm 0.54}$ | $84.3_{\pm 0.08}$ | $98.1_{\pm 0.04}$ | $81.6_{\pm 0.48}$ | $97.1_{\pm 0.08}$ | $86.0_{\pm 0.46}$ | $77.9_{\pm 0.49}$ |
| Spher. | $95.0_{\pm 0.41}$ | $88.1_{\pm 0.82}$ | $87.2_{\pm 0.04}$ | $84.1_{\pm 0.75}$ | $84.6_{\pm 0.12}$ | $98.3_{\pm 0.05}$ | $83.2_{\pm 0.29}$ | $98.1_{\pm 0.58}$ | $86.6_{\pm 0.01}$ | $72.1_{\pm 0.80}$ |
| Boot. | $86.6_{\pm 0.56}$ | $71.2_{\pm 1.17}$ | $82.0_{\pm 0.61}$ | $73.4_{\pm 1.06}$ | $80.5_{\pm 0.24}$ | $96.7_{\pm 0.06}$ | $77.3_{\pm 0.98}$ | $95.0_{\pm 0.25}$ | $77.0_{\pm 1.57}$ | $58.2_{\pm 2.99}$ |
| Trunc. | $97.1_{\pm 0.12}$ | $94.2_{\pm 0.39}$ | $87.8_{\pm 0.29}$ | $85.3_{\pm 0.77}$ | $84.1_{\pm 1.03}$ | $97.4_{\pm 1.03}$ | $83.1_{\pm 0.55}$ | $97.2_{\pm 1.00}$ | $88.3_{\pm 0.56}$ | $84.2_{\pm 0.69}$ |
| CL | $82.7_{\pm 0.57}$ | $67.5_{\pm 1.83}$ | $81.2_{\pm 0.34}$ | $73.1_{\pm 0.66}$ | $79.6_{\pm 0.17}$ | $96.4_{\pm 0.05}$ | $75.1_{\pm 0.67}$ | $94.2_{\pm 0.24}$ | $76.0_{\pm 2.16}$ | $59.4_{\pm 4.20}$ |
| ELR | $\mathbf{98.1}_{\pm 0.04}$ | $\mathbf{97.8}_{\pm 0.07}$ | $85.3_{\pm 0.23}$ | $83.4_{\pm 0.02}$ | $81.8_{\pm 0.26}$ | $97.5_{\pm 0.21}$ | $76.6_{\pm 0.10}$ | $96.5_{\pm 0.11}$ | $88.1_{\pm 0.82}$ | $85.7_{\pm 0.06}$ |
| FCE. | $95.4_{\pm 0.25}$ | $92.3_{\pm 0.13}$ | $83.6_{\pm 0.11}$ | $79.9_{\pm 0.78}$ | $83.1_{\pm 0.12}$ | $98.4_{\pm 0.20}$ | $80.6_{\pm 0.12}$ | $98.0_{\pm 0.03}$ | $84.7_{\pm 0.40}$ | $75.1_{\pm 0.04}$ |
| FCE+B | $95.7_{\pm 0.18}$ | $92.7_{\pm 0.74}$ | $84.8_{\pm 0.26}$ | $81.7_{\pm 0.27}$ | $83.4_{\pm 0.09}$ | $\mathbf{98.5}_{\pm 0.03}$ | $81.6_{\pm 0.51}$ | $\mathbf{98.1}_{\pm 0.15}$ | $86.7_{\pm 0.21}$ | $82.2_{\pm 0.06}$ |
| GCE | $94.4_{\pm 0.36}$ | $83.8_{\pm 1.14}$ | $86.4_{\pm 0.24}$ | $81.6_{\pm 0.37}$ | $84.3_{\pm 0.13}$ | $98.4_{\pm 0.08}$ | $82.7_{\pm 0.07}$ | $97.9_{\pm 0.02}$ | $81.1_{\pm 0.72}$ | $60.0_{\pm 1.31}$ |
| GCE+B | $96.6_{\pm 0.22}$ | $94.0_{\pm 0.13}$ | $86.5_{\pm 0.56}$ | $85.5_{\pm 0.13}$ | $84.1_{\pm 0.29}$ | $98.4_{\pm 0.04}$ | $82.8_{\pm 0.28}$ | $98.0_{\pm 0.06}$ | $86.1_{\pm 0.22}$ | $79.0_{\pm 1.17}$ |
| SCE | $89.5_{\pm 5.29}$ | $70.2_{\pm 0.69}$ | $82.7_{\pm 0.64}$ | $74.4_{\pm 0.37}$ | $82.1_{\pm 0.33}$ | $96.8_{\pm 0.10}$ | $79.6_{\pm 0.61}$ | $95.4_{\pm 0.15}$ | $78.2_{\pm 0.42}$ | $59.0_{\pm 4.43}$ |
| SCE+B | $97.0_{\pm 0.16}$ | $93.4_{\pm 0.29}$ | $87.5_{\pm 0.22}$ | $85.2_{\pm 0.98}$ | $83.5_{\pm 0.29}$ | $97.3_{\pm 0.14}$ | $81.8_{\pm 0.52}$ | $96.4_{\pm 0.20}$ | $88.9_{\pm 0.44}$ | $84.7_{\pm 0.37}$ |
| CE | $80.8_{\pm 2.31}$ | $67.3_{\pm 0.80}$ | $80.9_{\pm 1.11}$ | $72.1_{\pm 2.16}$ | $79.9_{\pm 0.28}$ | $96.4_{\pm 0.08}$ | $75.6_{\pm 0.20}$ | $94.2_{\pm 0.24}$ | $76.9_{\pm 1.22}$ | $59.9_{\pm 2.15}$ |
| CE+B | $96.2_{\pm 0.32}$ | $93.0_{\pm 0.09}$ | $87.9_{\pm 0.10}$ | $84.7_{\pm 0.37}$ | $80.8_{\pm 0.08}$ | $97.0_{\pm 0.04}$ | $78.9_{\pm 0.12}$ | $96.1_{\pm 0.26}$ | $84.5_{\pm 0.73}$ | $76.0_{\pm 1.13}$ |
| CEP | $97.5_{\pm 0.08}$ | $92.1_{\pm 0.44}$ | $87.8_{\pm 0.12}$ | $84.8_{\pm 0.23}$ | $85.5_{\pm 0.10}$ | $98.1_{\pm 0.07}$ | $84.3_{\pm 0.22}$ | $97.6_{\pm 0.14}$ | $84.2_{\pm 0.51}$ | $58.2_{\pm 2.94}$ |
| CEP+B | $95.6_{\pm 0.32}$ | $85.5_{\pm 0.77}$ | $\mathbf{88.1}_{\pm 0.31}$ | $84.2_{\pm 0.33}$ | $\mathbf{85.8}_{\pm 0.12}$ | $98.3_{\pm 0.02}$ | $\mathbf{84.8}_{\pm 0.10}$ | $98.0_{\pm 0.04}$ | $88.5_{\pm 0.32}$ | $85.1_{\pm 0.20}$ |

Table 2:  Test accuracies obtained using different losses on the noisy MNIST/ FashionM-NIST/EMNIST/CIFAR10 datasets. Losses implementing the noise-bound shaded in blue. When using this bound provides benefit, the corresponding value is $\boxed{boxed}$. Overall top values in **bold**.

## 6 Conclusion, Limitations and Further Work

In this work, we have looked at mitigating the impact of label noise in forward-corrected losses by training subject to a bound, motivated by our observation that label noise implies a minimum achievable risk.

**Summary**  We began by defining a family of loss functions we called 'generalised forward-corrected losses' since they contain correction losses as a strict subset. We showed how some popular existing robust losses can be formulated as generalised forward-corrected loss functions. We explained how label noise implies the existence of a lower bound on the achievable risk. We proposed training a model and preventing the

| Losses | CIFAR100 | | | | ASYM-CIFAR100 | | | | Non-Uniform-EMNIST | |
|---|---|---|---|---|---|---|---|---|---|---|
| | 0.2 | | 0.4 | | 0.2 | | 0.4 | | 0.6 | |
| | Top1 | Top5 | Top1 | Top5 | Top1 | Top5 | Top1 | Top5 | Top 1 | Top 5 |
| MSE | $57.2_{\pm0.93}$ | $78.6_{\pm0.25}$ | $40.6_{\pm0.38}$ | $63.0_{\pm0.24}$ | $56.3_{\pm0.11}$ | $82.6_{\pm0.22}$ | $40.7_{\pm0.12}$ | $74.4_{\pm0.25}$ | $44.7_{\pm2.66}$ | $86.7_{\pm3.10}$ |
| MAE | $10.0_{\pm0.11}$ | $13.8_{\pm0.28}$ | $7.6_{\pm1.89}$ | $11.6_{\pm1.25}$ | $7.1_{\pm6.02}$ | $11.1_{\pm6.6}$ | $11.1_{\pm5.43}$ | $25.1_{\pm5.76}$ | $9.8_{\pm1.74}$ | $23.1_{\pm1.80}$ |
| NCE | $38.7_{\pm3.13}$ | $51.8_{\pm3.77}$ | $19.1_{\pm0.20}$ | $28.8_{\pm0.15}$ | $16.3_{\pm1.24}$ | $25.4_{\pm1.80}$ | $21.8_{\pm1.24}$ | $37.2_{\pm1.80}$ | $18.0_{\pm1.17}$ | $38.8_{\pm1.93}$ |
| MixUp | $59.6_{\pm0.31}$ | $81.5_{\pm0.39}$ | $51.3_{\pm8.63}$ | $75.8_{\pm8.09}$ | $61.2_{\pm0.88}$ | $86.0_{\pm1.12}$ | $47.2_{\pm0.60}$ | $81.3_{\pm0.23}$ | $\mathbf{52.4}_{\pm0.80}$ | $\mathbf{95.5}_{\pm0.08}$ |
| Spher. | $57.7_{\pm0.18}$ | $82.9_{\pm0.54}$ | $48.8_{\pm0.51}$ | $74.3_{\pm0.73}$ | $54.2_{\pm0.32}$ | $81.2_{\pm0.29}$ | $39.2_{\pm0.31}$ | $72.1_{\pm0.15}$ | $41.9_{\pm0.10}$ | $94.4_{\pm0.04}$ |
| Boot. | $54.0_{\pm0.37}$ | $76.4_{\pm0.39}$ | $37.7_{\pm0.89}$ | $60.9_{\pm1.52}$ | $56.0_{\pm0.34}$ | $83.8_{\pm0.03}$ | $43.2_{\pm0.35}$ | $78.3_{\pm0.20}$ | $49.1_{\pm0.29}$ | $95.3_{\pm0.42}$ |
| Trunc. | $58.1_{\pm0.36}$ | $82.7_{\pm0.37}$ | $50.9_{\pm1.17}$ | $77.2_{\pm0.59}$ | $56.3_{\pm0.62}$ | $82.3_{\pm0.61}$ | $45.2_{\pm0.81}$ | $75.6_{\pm0.29}$ | $23.7_{\pm0.98}$ | $40.1_{\pm1.24}$ |
| CL | $53.0_{\pm0.21}$ | $76.3_{\pm0.19}$ | $36.3_{\pm0.77}$ | $60.1_{\pm0.66}$ | $55.3_{\pm0.48}$ | $83.5_{\pm0.28}$ | $42.4_{\pm0.45}$ | $78.1_{\pm0.14}$ | $48.2_{\pm0.45}$ | $95.0_{\pm0.04}$ |
| ELR | $10.4_{\pm0.24}$ | $31.7_{\pm0.44}$ | $10.0_{\pm0.64}$ | $30.1_{\pm0.88}$ | $10.8_{\pm0.21}$ | $32.7_{\pm0.53}$ | $10.3_{\pm0.39}$ | $30.8_{\pm0.35}$ | $40.3_{\pm0.39}$ | $93.0_{\pm0.24}$ |
| FCE | $56.9_{\pm0.58}$ | $79.2_{\pm0.14}$ | $43.7_{\pm0.15}$ | $66.2_{\pm0.19}$ | $55.3_{\pm0.54}$ | $83.5_{\pm0.24}$ | $41.4_{\pm0.55}$ | $77.3_{\pm0.75}$ | $39.0_{\pm0.05}$ | $67.8_{\pm0.47}$ |
| FCE+B | $56.1_{\pm2.22}$ | $81.8_{\pm1.37}$ | $50.2_{\pm0.02}$ | $77.2_{\pm0.19}$ | $54.2_{\pm0.44}$ | $83.3_{\pm0.43}$ | $43.8_{\pm0.02}$ | $77.5_{\pm0.13}$ | $40.0_{\pm0.35}$ | $73.2_{\pm0.08}$ |
| GCE | $60.0_{\pm0.13}$ | $82.6_{\pm0.63}$ | $44.9_{\pm0.07}$ | $67.2_{\pm0.34}$ | $53.8_{\pm0.55}$ | $81.6_{\pm0.14}$ | $39.4_{\pm0.44}$ | $74.0_{\pm0.36}$ | $44.8_{\pm0.62}$ | $91.2_{\pm0.70}$ |
| GCE+B | $59.4_{\pm0.02}$ | $83.5_{\pm0.24}$ | $50.3_{\pm0.11}$ | $75.3_{\pm0.64}$ | $55.4_{\pm0.55}$ | $83.0_{\pm0.35}$ | $46.5_{\pm1.44}$ | $77.7_{\pm0.35}$ | $47.1_{\pm0.20}$ | $93.5_{\pm0.43}$ |
| SCE | $55.9_{\pm0.53}$ | $76.5_{\pm0.15}$ | $38.7_{\pm0.60}$ | $60.9_{\pm0.41}$ | $57.5_{\pm0.19}$ | $83.7_{\pm0.17}$ | $43.3_{\pm0.87}$ | $77.5_{\pm0.75}$ | $47.2_{\pm0.33}$ | $92.5_{\pm0.01}$ |
| SCE+B | $55.5_{\pm0.90}$ | $77.4_{\pm0.84}$ | $47.1_{\pm1.32}$ | $69.2_{\pm1.18}$ | $57.9_{\pm0.83}$ | $83.7_{\pm0.41}$ | $50.0_{\pm1.62}$ | $80.4_{\pm0.65}$ | $47.9_{\pm0.80}$ | $93.8_{\pm0.05}$ |
| CE | $52.3_{\pm1.35}$ | $75.6_{\pm0.93}$ | $35.3_{\pm1.14}$ | $59.3_{\pm0.81}$ | $54.9_{\pm0.12}$ | $83.3_{\pm0.25}$ | $42.4_{\pm0.16}$ | $78.9_{\pm0.56}$ | $48.6_{\pm0.11}$ | $95.3_{\pm0.10}$ |
| CE+B | $50.9_{\pm1.01}$ | $76.5_{\pm0.86}$ | $39.9_{\pm1.02}$ | $65.8_{\pm1.19}$ | $52.9_{\pm1.86}$ | $83.2_{\pm0.88}$ | $34.7_{\pm2.51}$ | $73.4_{\pm1.50}$ | $45.5_{\pm5.11}$ | $93.0_{\pm0.16}$ |
| CEP | $58.8_{\pm0.87}$ | $78.6_{\pm0.38}$ | $43.5_{\pm0.24}$ | $65.1_{\pm1.27}$ | $59.4_{\pm0.08}$ | $82.2_{\pm0.03}$ | $46.5_{\pm0.17}$ | $76.4_{\pm0.25}$ | $48.2_{\pm0.05}$ | $95.4_{\pm0.07}$ |
| CEP+B | $\mathbf{62.3}_{\pm0.87}$ | $\mathbf{85.1}_{\pm0.46}$ | $\mathbf{54.3}_{\pm0.86}$ | $\mathbf{79.2}_{\pm0.93}$ | $\mathbf{63.0}_{\pm0.92}$ | $\mathbf{87.5}_{\pm0.32}$ | $\mathbf{53.0}_{\pm0.28}$ | $\mathbf{82.8}_{\pm0.13}$ | $45.0_{\pm0.48}$ | $95.0_{\pm0.08}$ |

Table 3: Test accuracies for different losses on the noisy CIFAR100/Asym-CIFAR100/Non-Uniform EMNIST datasets. Losses implementing the noise-bound shaded in blue. When using this bound provides benefit, the corresponding value is $\boxed{boxed}$. Overall top values in **bold**.

training loss going below a given threshold - we called this a 'bounded loss'. We derived this lower bound for generalised forward-corrected losses, showing it is the average entropy of the noisy label distribution (with respect to the entropy function of the base loss). We showed that uniform symmetric label noise is a 'worst-case' noise meaning that it has the highest entropy for a given noise rate $\eta$. When the label noise rate is known, but the noise model is otherwise unknown, we proposed using this worst-case entropy as a bound for our bounded loss. Finally, we empirically showed that training using the 'noise-bound' improves performance for different loss functions across various noisy settings.

### 6.1 Limitations and Future Work

While effective in specific settings, our method has limitations due to its reliance on a data separability assumption. This can restrict its effectiveness on datasets with inherent randomness. Future research could extend these methods to non-separable datasets. Also, while our approach improves on methods requiring detailed noise models, it can be applied only in settings where the noise rate is approximately known.

Although our proposed method generally offers benefits, there are observable differences in performance between different loss functions. Understanding these differences is a crucial direction for future research. Another promising area of future work involves extending these ideas to backward-corrections (Patrini et al., 2017), which are more prone to overfitting than forward-corrected losses.

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

# A Notation and Terminology

Table 4: Notation Table: Table summarising the notation used in this study.

| Symbol | Description |
|---|---|
| $c$ | Number of classes/labels. |
| $\mathcal{X}$ | The data domain, a subset of $\mathbb{R}^d$. |
| $\mathcal{Y}$ | The label space, defined as $1, 2, 3, \ldots, c$. |
| $\Delta$ | Probability simplex: The set of vectors $(p_1, p_2, \ldots, p_c)$ where each $p_i \geq 0$ and $\sum p_i = 1$. |
| $\boldsymbol{q}$ | A probability vector representing a forecast. |
| $\boldsymbol{p}$ | A probability vector representing ground-truth probabilities. |
| $\boldsymbol{q} : \mathcal{X} \to \Delta$ | A probability estimator model producing a forecast at each point in $\mathcal{X}$. |
| $\boldsymbol{p}(y \mid x)$ | The vector representing the conditional class probabilities at $x$, expressed as $\boldsymbol{p}(y \mid x) = (p(y = 1 \mid x), p(y = 2 \mid x), \ldots, p(y = c \mid x))$. |
| $f$ | A classifier function mapping each point in $\mathcal{X}$ to a label in $\mathcal{Y}$. |
| $L$ | The loss function used to evaluate the accuracy of predictions against actual labels. |
| $\boldsymbol{L}$ | The vector-valued function of the loss function $L$, where $\boldsymbol{L}(\boldsymbol{q}) = (L(\boldsymbol{q}, 1), \ldots, L(\boldsymbol{q}, c))$. |
| $R_L(\boldsymbol{q})$ | The $L$-risk of an estimator $\boldsymbol{q}$. |
| $R_L(\boldsymbol{q})(x)$ | The *pointwise* $L$-risk of an estimator $\boldsymbol{q}$ at $x$. |
| $R_L^\eta(\boldsymbol{q})$ | The noisy $L$-risk of an estimator $\boldsymbol{q}$. |
| $\mathcal{H}$ or $\mathcal{H}_L$ | The entropy function corresponding to the loss function $L$. |
| $H_L(\boldsymbol{p}, \boldsymbol{q})$ | The expected $L$-loss for a forecast $\boldsymbol{q}$ given the true label distribution $\boldsymbol{p}$. |
| $\eta$ | The noise rate of the label noise model. |
| $y, \widetilde{y}$ | The actual label and the noisy label, respectively. |
| $p(x, y)$ | The joint distribution of data and labels. |
| $\widetilde{p}(x, y)$ | The joint distribution of data and labels after corruption by label noise. |
| $p(\widetilde{y} \mid y, x)$ | The noise model generating noisy labels $\widetilde{y}$ from clean labels $y$ given $x$. |
| $T$ | The label noise transition matrix describing the probabilities of transforming a true label into a noisy label. |
| $\boldsymbol{e}_k$ | The standard basis vector in $\mathbb{R}^c$ where only the $k^{\text{th}}$ element is 1, and all others are 0. |

## A.1 Terminology

The majority of terminology which we adopt is the same as that found in any other contemporary machine learning paper on label noise. Label noise is categorised into two primary types: closed-set and open-set. We specifically address *closed-set label noise*, wherein the original label set and the noisy label set are identical. This is in contrast to open-set noise where the true label may not be included in the established label set (Wei et al., 2021). For instance, in a web-scraped dataset of animal images, a photograph of Tiger Woods' might be erroneously labeled as Tiger', even though the correct label, 'golfer', is absent from the set of labels When a label noise model has no dependence on the datapoint $p(\widetilde{y} \mid y, x) = p(\widetilde{y} \mid y)$ we say that it is 'uniform' or 'class-conditional', otherwise we sat that the label noise model is non-uniform. A uniform label noise model is called 'symmetric' where the transition probability between any distinct classes is the same and 'asymmetric' otherwise. Pairwise label noise is a subset of asymmetric label noise where mislabelling occurs between specific class pairs.

# B  Related Work

**Corruption identification methods:** Methods in this class first identify different types of corrupted samples and then re-weight, refine or remove these from the dataset. Many such methods rely on the heuristic that noisy samples have higher losses, especially earlier in training. This is based upon the well-known observation that complex models generally learn to classify easier data points before over-fitting on noise (Arpit et al., 2017). Song et al. (2019) use the entropy of the historical prediction distribution to identify refurbishable samples. Arazo et al. (2019) deploy a beta mixture model in the loss space and use the posterior probabilities that a sample is corrupted in the parameters of a bootstrapping loss. Zhou et al. (2020) define a loss which ignores samples that incur a higher loss value. Other approaches include a two-network model (Li et al., 2020) in which a Gaussian mixture model selects clean samples based on their loss values. These samples are then taken and used to train the other network. A number of other two-network models work on similar lines. Co-teaching (Han et al., 2018) trains two networks, one on the outputs of the other with the lowest loss values. Decoupling (Malach and Shalev-Shwartz, 2017) has the two networks update on the basis of disagreement with each other. Mentor-Net (Jiang et al., 2018) harnesses a teacher network for training a student network by re-weighting probably correct samples.

Zheng et al. (2019) is a meta-learning approach in which a label correction network corrects labels and feeds them to a classifier to train. This is done so that the performance of the classifier is optimised on a held-out validation set. Other meta-learning approaches include Ren et al. (2018) in which samples are re-weighted so that the learned classifier generalises better to a held-out clean meta-set. Similarly, in Vyas et al. (2020) (soft), labels are treated as learnable parameters and learned to maximise the performance on the meta-set. Other corruption identification methods expect that noisily labelled data lie heavily out of class distribution under an appropriate metric. In FINE (Kim et al., 2021), noisy samples are detected and removed using an eigendecomposition in the latent space. Alternatively, a KNN Feng et al. (2021a) in the latent space can identify and select samples based on their coherence to their neighbours' classes.

## B.1  Robust Loss Functions

An important set of methods for learning in the presence of noisy labels are robust loss methods. These methods work by substituting the cross-entropy objective for a loss which is less prone to inducing overfitting in the presence of label noise. An advantage here is the simplicity of these methods, as they do not require multiple networks or complex noise detection pipelines. This makes them suitable for plug-and-play use in any setting. These approaches may, crudely, be broken down into two classes; regularisation-based loss and correction-based losses.

**Correction-Based Losses:** Correction-based loss functions are motivated by the observation that label noise causes a distortion of the risk objective which can prevent consistency/Bayes-consistency. Methods in this class achieve robustness by altering the loss function to correct this distortion. We further subdivide these into correction-based losses and noise-tolerant losses. The former corrects the loss to compensate for the noising procedure (Larsen et al., 1998; Mnih and Hinton, 2012; Hendrycks et al., 2018a; Stempfel and Ralaivola, 2009). This procedure involves using noisy (Patrini et al., 2017) or clean data (Hendrycks et al., 2018b) to infer the noise transition matrix. The estimated noise model is then used to noise the model outputs, in the case of the 'forward-correction', or denoise the labels, in the case of the 'backward-correction' (Stempfel and Ralaivola, 2009; Patrini et al., 2017). A downside of these methods is the difficulty in estimating a noising matrix for a large number of classes and difficulty handling non-uniform noise models which impacts generality (Goldberger and Ben-Reuven, 2022; Sukhbaatar et al., 2015). Noise-tolerant loss functions (Ghosh and Kumar, 2017; Manwani and Sastry, 2013; Van Rooyen et al., 2015; Ma et al., 2020) use loss functions which ensure Bayes-consistency despite the presence of label noise, without the need to apply a correction to the loss function.

**Regularisation-Based Robust Losses:** Regularisation-based robust loss functions regularise the loss so that it less susceptible to overfitting to noise than the commonly used cross-entropy (Janocha and Czarnecki, 2016). Regularisation-based loss functions are varied. Many methods apply regularisation to ensure consistency between the predicted labels of nearby datapoints (Zhang et al., 2017; Englesson and Azizpour, 2021a;b; Iscen et al., 2022) or consistency of model predictions over time (Liu et al., 2020; Cheng

et al., 2024). Janocha and Czarnecki (2016) observe that $L^p$-losses typically used for regression show good robustness in a classification setting. This is particularly true for the MAE loss (Mean Absolute Error), which exhibits good robustness albeit with a tendency to under-fit and train slowly (Ma et al., 2020). This observation motivates a set of methods which combine or interpolate between CE and MAE to obtain the best of both. Wang et al. (2019) propose a solution to this by adding a 'reverse cross-entropy' (RCE) term to the usual cross-entropy (CE) term. Feng et al. (2021b) curtail the Taylor expansion of cross-entropy making it perform more like MAE. Generalised Cross-Entropy (Zhang and Sabuncu, 2018) construct a family of losses which interpolate between CE and MAE using a Box-Cox transformation in order to get the best of both. Other methods soften of mix labels to avoid overfitting (Reed et al., 2014; Szegedy et al., 2016; Thiel, 2008). Ishida et al. (2020), in-line with our work, bound the loss to prevent overfitting. However, their method is only briefly discussed in relation to label noise and provides no mechanism for selecting a loss bound. We ground this work firmly in the context of label noise and provide theoretical results for producing bounds on the loss.

## B.2 Algorithm

---

**Algorithm 1** Training with Noise-Bounded Loss

---

1: **Input:** Noisy dataset $\mathcal{D} = \{(x_i, \widetilde{y}_i)\}_{i=1}^N$, estimated noise rate $\eta$, number of classes $c$, epochs $T$
2: **Output:** Trained model parameters $\Theta$
3: **function** COMPUTENOISEBOUND($\eta$, $c$)
4: $\quad \boldsymbol{u}_{\text{sym}}(\eta, c) \leftarrow \left(1 - \eta, \frac{\eta}{c-1}, \ldots, \frac{\eta}{c-1}\right)$
5: $\quad$ **return** $\boldsymbol{u}_{\text{sym}}(\eta, c) \cdot L(\boldsymbol{u}_{\text{sym}}(\eta, c))$
6: **end function**
7: **procedure** TRAINMODEL($\mathcal{D}$, $\eta$, $c$, $T$)
8: $\quad B(\eta, c) \leftarrow$ COMPUTENOISEBOUND($\eta, c$)
9: $\quad$ **for** $epoch = 1$ to $T$ **do**
10: $\quad\quad$ **for** each $(x_i, y_i)$ in $\mathcal{D}$ **do**
11: $\quad\quad\quad \boldsymbol{q}(x_i) \leftarrow$ ModelPrediction($x_i; \Theta$)
12: $\quad\quad\quad loss \leftarrow \left\| B(\eta, c) - \frac{1}{N} \sum_{j=1}^N L(\boldsymbol{q}(x_j), y_j) \right\|_1$
13: $\quad\quad\quad \Theta \leftarrow$ UpdateModel($\Theta, loss$)
14: $\quad\quad$ **end for**
15: $\quad$ **end for**
16: $\quad$ **return** $\Theta$
17: **end procedure**

---

# C Proofs

The following standard result regarding proper losses due to Savage (Savage, 1971; Gneiting and Raftery, 2007) is indispensable in subsequent proofs.

**Theorem C.1** (Savage's Theorem). *A differentiable loss function L is (strictly) proper if and only if there exists a (strictly) concave function $\mathcal{J} : \mathbb{R}^c \to \mathbb{R}$ such that for each $\boldsymbol{q} \in \Delta$ and $k \in \mathcal{Y}$,*

$$L(\boldsymbol{q}, k) = \nabla \mathcal{J}(\boldsymbol{q})(\boldsymbol{e}_k - \boldsymbol{q}) + \mathcal{J}(\boldsymbol{q}).$$

*Moreover, $\mathcal{J}$ is precisely the entropy function for L. Thus, in particular, a loss is (strictly) proper if and only if its associated entropy function is (strictly) concave.*

**Definition C.2** (Generalised Forward-Correction). Let $L_f$ be a loss function and $f : \Delta \to \Delta$ be an injective function. We say $L_f$ is a 'generalised forward-corrected loss' if there exists a loss function $L$ such that for all $\boldsymbol{q} \in \Delta$, $k \in \{1, 2, \ldots, c\}$

$$L_f(\boldsymbol{q}, k) = L(f(\boldsymbol{q}), k)$$

We refer to $L$ as the **base loss**. $f$ can be thought of as a label noise model.

**Lemma C.3.** The GCE, SCE and FCE losses can be formulated as generalised forward-correction losses with a proper base loss. The noise models $f_{GCE}, f_{SCE}, f_{FCE}$ satisfy

$$(f_{GCE}^{-1}(\boldsymbol{p}))_i = \frac{p_i^{\frac{1}{1-a}}}{\sum_{i=1}^c p_i^{\frac{1}{1-a}}},$$

$$(f_{SCE}^{-1}(\boldsymbol{p}))_i = \frac{p_i}{\lambda - Ap_i},$$

$$f_{FCE}(\boldsymbol{p}) = T^{-1}\boldsymbol{p},$$

where $T$ is the invertible stochastic matrix used to define the correction, and $\lambda$ is a constant selected to ensure the correct normalisation.

**Proof Idea:** Suppose that $L$ is a proper loss, let $f : \Delta \to \Delta$ be injective noise-model, and consider the minimiser of the expected loss defined $L_f(\boldsymbol{q}, k) := L(f(\boldsymbol{q}), k)$ at $\boldsymbol{p} \in \Delta$;

$$\underset{\boldsymbol{q}\in\Delta}{\arg\min}\, H(\boldsymbol{p}, \boldsymbol{q}) = \underset{\boldsymbol{q}\in\Delta}{\arg\min} \sum_{i=1}^c p_i L_f(\boldsymbol{q}, i)$$

$$= \underset{\boldsymbol{q}\in\Delta}{\arg\min} \sum_{i=1}^c p_i L(f(\boldsymbol{q}), i).$$

Since $L$ is proper then we know this is minimised by $\boldsymbol{q}$ such that $f(\boldsymbol{q}) = \boldsymbol{p}$. In other words *we can uncover the noise model $f$ by finding the minimiser of the expected loss.* This is how we find $f$ for each of the loss functions. The core idea of the following proof is to write out the expected loss for each loss function and, for each $\boldsymbol{p} \in \Delta$, to find $\arg\min_{\boldsymbol{q}\in\Delta} H(\boldsymbol{p}, \boldsymbol{q})$. Assuming that this $\arg\min$ consists of a single point then this induces a map $f(\boldsymbol{p}) := \arg\min_{\boldsymbol{q}\in\Delta} H(\boldsymbol{p}, \boldsymbol{q})$ which, for the reasons given, can be identified with the noise model.

*Proof.* We begin by introducing the following notation: Let $L$ be an elementwise loss and let $\boldsymbol{p}, \boldsymbol{q}$ be two distributions, we denote the expected loss of $\boldsymbol{q}$ with respect to $\boldsymbol{p}$ to be $H_L(\boldsymbol{q}, \boldsymbol{p}) := \sum_{i=1}^c p_i L(\boldsymbol{q}, i)$.

Let us begin by considering GCE. The expected loss may be written $L_{GCE}(\boldsymbol{q}, \boldsymbol{p}) := \sum_{i=1}^c p_i L_{GCE}(\boldsymbol{q}, i) := \sum_{i=1}^c p_i \frac{1-q_i^a}{a}$. We find the minima by constructing the Langrangian $A(\boldsymbol{q}, \lambda) := \sum_{i=1}^c p_i \frac{1-q_i^a}{a} + \lambda(\sum_{i=1}^c q_i - 1)$. By taking partials and equating to zero, we obtain $q_i^{1-a} = \frac{ap_i}{\lambda}, \forall i$. Using the fact that $\sum_{i=1}^c q_i = 1$ one may find the value of $\lambda$. Specifically, $\lambda = a(\sum_{i=1}^c p_i^{\frac{1}{1-a}})^{1-a}$. Thus overall one has $q_i^* = (\frac{ap_i}{\lambda})^{\frac{1}{1-a}} = \frac{p_i^{\frac{1}{1-a}}}{\sum_{i=1}^c p_i^{\frac{1}{1-a}}}$.

Let us repeat this for the SCE loss. The expected loss may be written $L_{SCE}(\boldsymbol{q}, \boldsymbol{p}) := \sum_{i=1}^c p_i L_{SCE}(\boldsymbol{q}, i) := \sum_{i=1}^c p_i(A(1 - q_i) - log(q_i))$. As before, we construct the relevant Lagrangian and find the stationary points: $B(\boldsymbol{q}, \lambda) := \sum_{i=1}^c p_i(A(1 - q_i) - log(q_i)) + \lambda(\sum_{i=1}^c q_i - 1)$. Taking partials and equating to zero we obtain $p_i(A + \frac{1}{q_i}) = \lambda \implies q_i^* = \frac{p_i}{\lambda - Ap_i}$. Here the value of the normalisation constant $\lambda$ cannot be found in closed form for high values of $c$ and must be computed numerically. Finally, we consider the forward-corrected CE loss. We assume that the loss is corrected by some invertible stochastic matrix $T$. $L_F(\boldsymbol{q}, \boldsymbol{p}) := \sum_{i=1}^c p_i L_F(\boldsymbol{q}, i) := \sum_{i=1}^c -p_i log((T\boldsymbol{q})_i)$. We remark that since CE is proper that this is minimised on the simplex by $\boldsymbol{p} = T\boldsymbol{q}^* \iff \boldsymbol{q}^* = T^{-1}\boldsymbol{p}$. For each loss, the function $f$ obtained is injective as desired. $\square$

## C.1 Entropy Bounds

**Lemma C.4.** Let $L_f$ be a generalised-correction-loss whose 'base-loss' $L$ is strictly proper (Recall the definition of 'base-loss' from Definition 2.1). The noisy risk of any probability estimator $\boldsymbol{q}$ is lower bounded:

$$R_{L_f}^\eta(\boldsymbol{q}) \geq \mathbb{E}_{x\sim p(x)}[\mathcal{H}(\widetilde{\boldsymbol{p}}(\widetilde{y}|x))], \tag{9}$$

where $\mathcal{H}$ is the entropy function of the base-loss. This bound is tight when $f$ is equal to the true noise model. Equality is attained by setting $\boldsymbol{q}(x) = f^{-1}(\widetilde{\boldsymbol{p}}(\widetilde{y}|x))$.[2]

---

[2] Note that if $f$ is the true noise model then $\widetilde{\boldsymbol{p}}(\widetilde{y} \mid x) \in f(\Delta)$ and the inverse is unique by injectivity.

*Proof.* Recollect that $L_f$ is a generalised forward-correction loss with (strictly) proper base loss $L$: $L_f(\boldsymbol{q}, k) = L(f(\boldsymbol{q}, k))$. Let $x$ be some arbitrary point in the support of $p(x)$ and let $\boldsymbol{q}(x)$ be some probability estimator. The pointwise noisy risk of $\boldsymbol{q}$ at $x$ may be written as

$$
\begin{aligned}
R^{\eta}_{L_f}(\boldsymbol{q})(x) &:= \sum_{i=1}^{c} \widetilde{p}(\widetilde{y} = i|x) L_f(\boldsymbol{q}(x), i) \\
&= \sum_{i=1}^{c} \widetilde{p}(\widetilde{y} = i|x) L(f(\boldsymbol{q}(x)), i) \\
&\geq \sum_{i=1}^{c} \widetilde{p}(\widetilde{y} = i|x) L(\widetilde{\boldsymbol{p}}(\widetilde{y}|x), i) \\
&=: \mathcal{H}(\widetilde{\boldsymbol{p}}(\widetilde{y}|x))
\end{aligned}
$$

The inequality follows from the definition of the properness of $L$. Inequality 4 follows by taking expectation with respect to $p(x)$ on both sides. Equality is attained setting by $f(\boldsymbol{q}(x)) = \widetilde{\boldsymbol{p}}(\widetilde{y}|x)$ for each $x$, (equivalently $f^{-1}(\boldsymbol{q}(x)) = \widetilde{\boldsymbol{p}}(\widetilde{y}|x)$) which is possible when $f$ is the true noise model as then $\widetilde{\boldsymbol{p}} \in f(\Delta)$. The injectivity of $f$ (as specified in the definition of $f-$proper) means this occurs uniquely at $\boldsymbol{q}(x) = f(\widetilde{\boldsymbol{p}}(\widetilde{y}|x))$ as desired. □

**Lemma C.5** (Class-Conditional Label Noise). When the classes are balanced and label noise is asymmetric and given by transition matrix $T$, the noisy risk of a probability estimator $\boldsymbol{q}$ may be lower bounded as follows,

$$
R^{\eta}_{L_f}(\boldsymbol{q}) \geq \frac{1}{c} \sum_{i=1}^{c} \mathcal{H}(\boldsymbol{T}_{\cdot,i}), \tag{10}
$$

where $\boldsymbol{T}_{\cdot,i}$ denotes the $i^{\text{th}}$ column of the matrix $T$.

*Proof.* The right-hand side of Inequality 4 can be written

$$
\begin{aligned}
\mathbb{E}_{x \sim p(x)} \left[ \mathcal{H}(\widetilde{\boldsymbol{p}}(\widetilde{y}|x)) \right] &= \mathbb{E}_{x \sim p(x)} \left[ \sum_{i=1}^{c} \widetilde{p}(\widetilde{y} = i|x) L(\widetilde{\boldsymbol{p}}(\widetilde{y}|x), i) \right] \\
&= \mathbb{E}_{x \sim p(x)} \left[ T\boldsymbol{p}(y|x) \cdot \boldsymbol{L}(T\boldsymbol{p}(y|x)) \right] \\
&= \frac{1}{c} \sum_{k=1}^{c} (T\boldsymbol{e}_k) \cdot \boldsymbol{L}(T\boldsymbol{e}_k),
\end{aligned}
$$

where the final equality comes from using the fact that classes are balanced and all points are anchor points. This is equal to

$$
\frac{1}{c} \sum_{k=1}^{c} T_{\cdot,k} \cdot \boldsymbol{L}(T_{\cdot,k}) = \frac{1}{c} \sum_{k=1}^{c} \mathcal{H}(\boldsymbol{T}_{\cdot,k}),
$$

as desired. □

**Corollary C.6** (Uniform Symmetric Label Noise). Given uniform, symmetric label noise at rate $\eta$, the risk associated with any probability estimator can be bounded as follows:

$$
R^{\eta}_{L_f}(\boldsymbol{q}) \geq \mathcal{H}\left(1 - \eta, \frac{\eta}{c-1}, \frac{\eta}{c-1}, \ldots, \frac{\eta}{c-1}\right). \tag{11}
$$

This can be written equivalently as

$$
R^{\eta}_{L_f}(\boldsymbol{q}) \geq \boldsymbol{u}_{\text{sym}}(\eta, c) \cdot \boldsymbol{L}(\boldsymbol{u}_{\text{sym}}(\eta, c)).
$$

where

$$
\boldsymbol{u}_{\text{sym}}(\eta, c) := \left(1 - \eta, \frac{\eta}{c-1}, \frac{\eta}{c-1}, \ldots, \frac{\eta}{c-1}\right). \tag{12}
$$

*Proof.* When label is symmetric every column of the matrix $T$ is a permutation of $\left(1 - \eta, \frac{\eta}{c-1}, \frac{\eta}{c-1}, \dots, \frac{\eta}{c-1}\right)$. The result follows immediately from the symmetry assumption on the entropy. □

**Lemma C.7** (Non-Uniform Symmetric Label Noise)**.** Let $p(x, y)$ be a separable distribution, and let $\widetilde{p}(x, \widetilde{y})$ be a noisy distribution obtained by applying non-uniform symmetric label noise to $p(x, y)$. Assume that $L$ is a generalised forward-correction losses loss and let $\mathcal{H}$ denote the (symmetric) entropy function of its base loss. For any probability estimator $\boldsymbol{q}$, we have the following lower bound on its noisy risk,

$$R_{L_f}^{\eta}(\boldsymbol{q}) \geq \mathbb{E}_{x \sim p(x)}\left[\mathcal{H}\left(1 - \eta(x), \frac{\eta(x)}{c - 1}, \frac{\eta(x)}{c - 1}, \dots, \frac{\eta(x)}{c - 1}\right)\right],$$

where $\eta(x)$ denotes the noise rate at $x$. This inequality is strict and may be obtained by setting $\boldsymbol{q}(x) = f^{-1}(\widetilde{\boldsymbol{p}}(y \mid x))$, if $\widetilde{\boldsymbol{p}}(y \mid x) \in f(\Delta)$.

*Proof.* The right-hand side of Inequality 4 can be written

$$\mathbb{E}_{x \sim p(x)}\left[\mathcal{H}(\widetilde{\boldsymbol{p}}(\tilde{y}|x))\right] = \mathbb{E}_{x \sim p(x)}\left[\sum_{i=1}^{c} \widetilde{p}(\widetilde{y} = i|x) L(\widetilde{\boldsymbol{p}}(\widetilde{y}|x), i)\right]$$

$$= \mathbb{E}_{x \sim p(x)}\left[T(x)\boldsymbol{p}(y|x) \cdot \boldsymbol{L}(T(x)\boldsymbol{p}(y|x))\right].$$

Our separability assumption means that $\boldsymbol{p}(y|x) = \boldsymbol{e}_k$ for some $k$. For each $x$ it follows that $T(x)\boldsymbol{p}(y \mid x)$ is some rearrangement of the vector $(1 - \eta(x), \frac{\eta(x)}{c-1}, \frac{\eta(x)}{c-1}, \dots, \frac{\eta(x)}{c-1})$. By the assumption that the entropy function is symmetric we may conclude that

$$\mathbb{E}_{x \sim p(x)}\left[\mathcal{H}(\widetilde{\boldsymbol{p}}(\tilde{y}|x))\right] = \mathbb{E}_{x \sim p(x)}\left[\mathcal{H}\left(1 - \eta(x), \frac{\eta(x)}{c - 1}, \frac{\eta(x)}{c - 1}, \dots, \frac{\eta(x)}{c - 1}\right)\right].$$

□

### C.1.1 The General Case

**Lemma C.8.** Lemma 4.2 establishes that we can lower bound the noisy risk of an estimator by the average entropy of the noisy conditional class distribution

$$R_{L_f}^{\eta}(\boldsymbol{q}) \geq \mathbb{E}_{x \sim p(x)}[\mathcal{H}(\widetilde{p}(\widetilde{y} \mid x)].$$

Given that the noise rate at $x$ is $\eta(x)$, this Lemma establishes that $\mathcal{H}(\widetilde{p}(\widetilde{y} \mid x)$ must lie in the following interval:

$$\mathcal{H}(\widetilde{p}(\widetilde{y} \mid x) \in \left[\mathcal{H}(1 - \eta(x), \eta(x), 0, 0, \dots, 0), \mathcal{H}\left(1 - \eta(x), \frac{\eta(x)}{c - 1}, \frac{\eta(x)}{c - 1}, \dots, \frac{\eta(x)}{c - 1}\right)\right].$$

In particular, given a fixed noise rate $\eta(x)$ at $x$, the highest possible entropy occurs when label noise is symmetric at $x$. The lowest entropy occurs when the label noise is pairwise.

*Proof.* Let $\boldsymbol{q}(x)$ be a probability estimator and let $x$ be some point in the support of $p(x)$. We established in the proof of Lemma 4.2 that $R_L^{\eta}(\boldsymbol{q})(x) \geq \mathcal{H}(\widetilde{\boldsymbol{p}}(\tilde{y}|x))$. We have equality (uniquely) when $\boldsymbol{q}(x) = \widetilde{\boldsymbol{p}}(\tilde{y}|x)$. Let $T(x)$ denote the noising transition matrix at $x$. By the separability assumption, we have some $k$ such that $p(y = k|x) = 1$ and $p(y = i|x) = 0$ otherwise. Thus $\tilde{p}(\tilde{y}|x) = \sum_{y=1}^{c} \tilde{p}(\tilde{y}|y, x) p(y|x) = \tilde{p}(\tilde{y}|y = k, x) = (T_{1k}(x), T_{2k}(x), \dots, T_{ck}(x))$. Let $A(\eta(x), c) := \mathcal{H}(T_{1k}(x), T_{2k}(x), \dots, T_{ck}(x))$ where $\eta(x) := 1 - T_{kk}$ is the noise rate at $x$. The symmetry of $\mathcal{H}$ means that, without loss of generality, we may let $k = 1$. It remains to show that $A(\eta(x), c) \in \left[\mathcal{H}(1 - \eta(x), \eta(x), 0, 0, \dots, 0), \mathcal{H}\left(1 - \eta(x), \frac{\eta(x)}{c-1}, \frac{\eta(x)}{c-1}, \dots, \frac{\eta(x)}{c-1}\right)\right]$.

**Upper Limit:** We begin by demonstrating that $A(\eta(x), c)$ is upper bounded by $\mathcal{H}(1 - \eta(x), \frac{\eta(x)}{c-1}, \frac{\eta(x)}{c-1}, \dots, \frac{\eta(x)}{c-1})$. Let $\Delta(\eta(x))$ denote the set of non-negative vectors $(a_1, a_2, \dots, a_{c-1})$ such that $a_i \leq 1$ and $\sum_{i=1}^{c-1} a_i = \eta(x)$. We

wish to show the supremum of $\mathcal{H}(1-\eta(x), a_1, a_2, \ldots, a_{c-1})$ is attained on $\Delta(\eta(x))$ by setting $a_i = \frac{\eta(x)}{c-1}$ for all $i$. This corresponds to the label noise being symmetric at $x$. By Theorem C.1 $\mathcal{H}$ is a (strictly) concave function. Moreover, the symmetry assumption implies that $\mathcal{H}$ is a symmetric function of its variables. Define the function $g(a_1, a_2, \ldots, a_{c-1}) := \mathcal{H}(1-\eta(x), a_1, a_2, \ldots, a_{c-1})$. We wish to show that $g$ attains its maximum on $\Delta(\eta(x))$ when $a_i = a_j$ for all $i, j$. We begin by noting that the (strict) concavity of $\mathcal{H}$ implies the (strict) concavity of $g$. To see this consider two arbitrary vectors $\boldsymbol{x} = (x_1, x_2, \ldots x_{c-1})$, $\boldsymbol{y} = (y_1, y_2, \ldots y_{c-1})$. Now $g(\lambda\boldsymbol{x}+(1-\lambda)\boldsymbol{y}) = \mathcal{H}(\lambda\boldsymbol{x}' + (1-\lambda)\boldsymbol{y}')$ where $\boldsymbol{x}' := (1-\eta(x), x_1, x_2, \ldots, x_{c-1})$ and $\boldsymbol{y}' := (1-\eta(x), y_1, y_2, \ldots, y_{c-1})$. Thus the concavity of $\mathcal{H}$ implies $g(\lambda\boldsymbol{x}+(1-\lambda)\boldsymbol{y}) := \mathcal{H}(\lambda\boldsymbol{x}'+(1-\lambda)\boldsymbol{y}') \geq \lambda\mathcal{H}(\boldsymbol{x}')+(1-\lambda)\mathcal{H}(\boldsymbol{y}') = \lambda g(\boldsymbol{x})+(1-\lambda)g(\boldsymbol{y})$ as desired. Thus $g$ is a symmetric (strictly) concave function of its variables.

Let $\boldsymbol{a}^*$ denote a maxima of $g$ on $\Delta(\eta(x))$. Let $\sigma$ denote the cyclic permutation of the components of $\boldsymbol{a}$. That is $\sigma(a_1, a_2, \ldots, a_{c-1}) := (a_{c-1}, a_1, a_2, \ldots, a_{c-2})$. By the symmetry of $g$, we know that if $\boldsymbol{a}^*$ is a maxima then so is $\sigma^i(\boldsymbol{a}^*)$ for all $i$: $g(\boldsymbol{a}^*) = g(\sigma^i(\boldsymbol{a}^*))$ for all $i \in \mathbb{N}$. The defining property of a concave function is that

$$g(\lambda_1 \boldsymbol{v}_1 + \lambda_2 \boldsymbol{v}_2 + \ldots + \lambda_d, \boldsymbol{v}_d) \geq \sum_{i=1}^{d} \lambda_i g(\boldsymbol{x}_i)$$

$$\text{where } \sum_i \lambda_i = 1.$$

Hence by the (strict) concavity of $g$, setting $\lambda_i := \frac{1}{c-1}$;

$$g\left(\frac{\eta(x)}{c-1}, \frac{\eta(x)}{c-1}, \ldots, \frac{\eta(x)}{c-1}\right) = g\left(\frac{1}{c-1}(\boldsymbol{a}^* + \sigma(\boldsymbol{a}^*) + \sigma^2(\boldsymbol{a}^*) + \ldots + \sigma^{c-2}(\boldsymbol{a}^*))\right)$$

$$\geq \frac{1}{c-1}g(\boldsymbol{a}^*) + \frac{1}{c-1}g(\sigma(\boldsymbol{a}^*)) + \ldots + \frac{1}{c-1}g(\sigma^{c-2}(\boldsymbol{a}^*))$$

$$= g(\boldsymbol{a}^*)$$

Hence $g$ is maximised by setting $a_i = \frac{\eta(x)}{c-1}$ for all $i$ as desired. This is the unique maxima when the base loss strictly proper.

**Lower Limit:** It now remains to show that the lower bound on $A(\eta(x), c)$ holds. The (strict) concavity means that $g$ attains it minima on the vertices of $\Delta(\eta(x))$ (eg $(\eta(x), 0, \ldots, 0)$. To see this let $\boldsymbol{a}^* = (a_1^*, a_2^*, \ldots, a_{c-1}^*)$ denote a minima of $g$ on $\Delta(\eta(x))$. Then we have,

$$g(a_1^*, a_2^*, \ldots, a_{c-1}^*) = g(a_1^* \boldsymbol{e_1} + a_2^* \boldsymbol{e_2} + \ldots + a_{c-1}^* \boldsymbol{e_{c-1}})$$

$$\geq \sum_{i=1}^{c-1} \frac{a_i^*}{\eta(x)} g(\eta(x)\boldsymbol{e_i})$$

$$= g(\eta(x), 0, \ldots, 0) \tag{13}$$

$$= \mathcal{H}(1-\eta(x), \eta(x), 0, 0, \ldots, 0)$$

$\boldsymbol{e}_i$ denotes the coordinate vector with 1 in the $i$th position and zeros elsewhere. Equation 13 holds by the symmetry of $g$ and since $\sum a_i^* = \eta(x)$. Thus we have shown that $g$ is lower bounded by $\mathcal{H}(1 - \eta(x), \eta(x), 0, 0, \ldots, 0)$ as desired. Moreover, this infimum is obtained on the vertices of $\Delta(\eta(x))$. $\square$

**Corollary C.9.** Given an average noise rate $\eta := \mathbb{E}_{x \sim p(x)}[\eta(x)]$, the greatest possible value of $\mathbb{E}_{x \sim p(x)}[\mathcal{H}(\widetilde{p}(\widetilde{y} \mid x)]$ occurs when $\eta(x)$ is constant:

$$\sup_{p(\widetilde{y}|x,y)} \left(\mathbb{E}_{x \sim p(x)}[\mathcal{H}(\widetilde{p}(\widetilde{y} \mid x))]\right) = \mathcal{H}\left(1-\eta, \frac{\eta}{c-1}, \frac{\eta}{c-1}, \ldots, \frac{\eta}{c-1}\right),$$

where the supremum is taken over all noise models such that $\mathbb{E}_{x \sim p(x)}[\eta(x)] = \eta$.

*Proof.* We established in the proof of Lemma 4.3 that, given that the noise rate at $x$ is $\eta(x)$,

$$\mathcal{H}(\widetilde{p}(\widetilde{y} \mid x) \in \left[\mathcal{H}(1-\eta(x), \eta(x), 0, 0, \ldots, 0), \mathcal{H}\left(1-\eta(x), \frac{\eta(x)}{c-1}, \frac{\eta(x)}{c-1}, \ldots, \frac{\eta(x)}{c-1}\right)\right].$$

Thus, given a fixed average noise rate $\eta$ we maximise the expected entropy when the noise model describes symmetric label noise at each point in dataspace. We now wish to demonstrate that

$$\mathbb{E}_{x \sim p(x)} \left[ \mathcal{H}\left(1 - \eta(x), \frac{\eta(x)}{c-1}, \frac{\eta(x)}{c-1}, \dots, \frac{\eta(x)}{c-1}\right) \right] \leq \mathcal{H}\left(1 - \eta, \frac{\eta}{c-1}, \frac{\eta}{c-1}, \dots, \frac{\eta}{c-1}\right),$$

which is to say that we maximise the entropy of symmetric noise by setting $\eta(x) = const$. $\mathcal{H}$ is concave (strictly concave if the base loss is strictly proper) thus we can use Jensen's Inequality which tells us that

$$f(\mathbb{E}[X]) \geq \mathbb{E}[f(X)],$$

if $f$ is concave. Hence, by setting our random variable $X := (1 - \eta(x), \eta(x)/(c-1), \dots, \eta(x)/(c-1))$, and $f = \mathcal{H}$ yields the desired result. $\qquad\square$

## D   Additional Theory and Discussion

### D.1   Example: Forward-Corrected CE

This section gives a breakdown of the example from Table 1. We demonstrate how a neural network classifier can achieve a training loss lower than any model that has not accessed the training labels, indicating that it must have overfit.

Consider a separable distribution $p(x, y)$ corrupted by uniform, symmetric label noise at a rate of 40%. Assume this noise model is known and corrected for in the loss calculation. We utilise a cross-entropy loss, hence the forward-corrected loss is defined:

$$L_F(q, 1) = -\log(0.6q + 0.4(1 - q)) = -\log(0.4 + 0.2q),$$
$$L_F(q, 0) = -\log(0.4q + 0.6(1 - q)) = -\log(0.6 - 0.2q).$$

We generate a large noisy dataset from $p(x, y)$, denoted $\widetilde{D} = \{(x_i, \widetilde{y}_i)\}_{i=1}^N$. Our objective is to construct a model that attains minimal loss on $\widetilde{D}$.

**Scenario One: No Label Peeking**   Without peeking at the dataset labels, we minimise the expected loss on the noisy dataset by setting our model predictions $q(x)$ such that, for every $x \sim p(x)$, $q(x)$ minimises the noisy expected loss with respect to $L_F$. For example, consider a point $x_0 \sim p(x)$ with a clean label $y_0 = 1$, implying $p(y_0 = 1 \mid x_0) = 1$. The noisy conditional class distribution at $x_0$ is $\widetilde{p}(\widetilde{y}_0 \mid x_0) = 0.6$. The noisy expected loss at $x_0$ is:

$$\begin{aligned}
H_{L_F}(\widetilde{p}, q) &= 0.6 L_F(q, 1) + 0.4 L_F(q, 0) \\
&= -0.6 \log(0.4 + 0.2q) - 0.4 \log(0.6 - 0.2q).
\end{aligned}$$

This loss is minimised by setting $q = 1$, yielding an expected noisy loss of 0.673 at $x_0$. Similarly, for $y = 0$, setting $q(x) = 0$ achieves the same loss.

**Scenario Two: Label Peeking**   Conversely, if a model can peek at the noisy dataset labels before forecasting, it minimises the empirical risk by setting $q(x) = 1$ if $\widetilde{y}_0 = 1$ and $q(x) = 0$ if $\widetilde{y}_0 = 0$, attaining a noisy risk of:

$$L_F(q = 1, 1) = L_F(q = 0, 0) = -\log(0.6) = 0.511.$$

Therefore, while an over-parameterised neural network model trained for sufficient epochs may achieve a training loss of 0.511, the lowest possible training loss for an optimal model without label access is 0.673.

| Loss Function | Optimal Training Loss | |
| --- | --- | --- |
| | Label Peeking | No Peeking |
| CE | 0 | 0.673 |
| FCE | 0.511 | 0.673 |
| FCE+B | 0.673 | 0.673 |

Table 5: Comparison of the minimum loss which may be achieved by a model on a (large) dataset when using different loss function both *allowing* and *not-allowing* peeking at the dataset labels assuming 40% symmetric label noise on a separable binary label dataset. The results highlight a significant overfitting potential in cross-entropy, as indicated by the large gap between the peeking and non-peeking scenarios. While FCE introduces an inherent loss bound, improving robustness, it may still permit overfitting. Our bounded variant, FCE+B, is designed to better align with the dataset and mitigate overfitting.

### D.2 When Will Noise-Bounding Be Effective?

When label noise is uniform and symmetric, the noise-bound (Definition 4.5) is equal to the entropy of the noisy label distribution. However, when label noise deviates from these idealised assumptions, the noise-bound is strictly greater than entropy of the noisy label distribution. Consequently, in asymmetric noise settings we should expect noise-bounding to be less effective than in the symmetric noise setting. In this section we analyze the effectiveness of noise-bounding for asymmetric noise models. We hypothesize that noise-bounding will be effective for asymmetric noise models when the underlying entropy function of the chosen loss function is *'insensitive'*. By 'insensitive' we mean that two distributions with the same noise *rate* will have a similar entropy, even if the noise models which generated these distribution differ greatly. We show the entropy functions for GCE and SCE are insensitive whereas the Jensen-Shannon entropy is sensitive. We predict from this that noise-bounding will less effective for cross-entropy for asymmetric label noise. This prediction is validated by our experiment in Section D.2.1 where we show that noise-bounding is effective for SCE, regardless of the noise model, whereas CE performs poorly for asymmetric noise.

The noise-bound will equal the entropy of the noisy label distribution only when label noise is uniform and symmetric. When label noise is *asymmetric* there is a gap between the noise-bound and the actual noisy entropy. Specifically, in the asymmetric noise setting, the noise bound will be strictly greater than the true entropy. In these settings we would expect to underfit. Crucially, the extent to which we will underfit is likely to be regulated by the size of this gap; underfitting being less pronounced if the gap between the noise bound and the entropy is small. Given a noise rate $\eta$, the following Lemma gives the worst-case gap between the true entropy of the noisy distribution and the noise-bound, assuming uniform label noise.

**Corollary D.1.** Suppose we have some uniform label noise at noise rate $\eta$. Let $\mathcal{H}$ denote the average entropy of the noisy label distribution, that is

$$\mathcal{H} := \mathbb{E}_{x \sim p(x)}[\mathcal{H}(\widetilde{\boldsymbol{p}}(y \mid x))].$$

Let $B(\eta, c)$ denote the noise-bound defined in Definition 4.5. Then

$$|B(\eta, c) - \mathcal{H}| \leq \mathcal{H}\left(1 - \eta, \frac{\eta}{c-1}, \frac{\eta}{c-1}, \ldots, \frac{\eta}{c-1}\right) - \mathcal{H}(1 - \eta, \eta, 0, 0, \ldots, 0)$$

*Proof.* This follows immediately from Lemma 4.3 when $\eta(x)$ has no dependence on $x$ ($\eta(x) = \eta$). □

As discussed previously, when noise is uniform but not symmetric, our noise-bound (Definition 4.5) of $\mathcal{H}(1 - \eta, \frac{\eta}{c-1}, \frac{\eta}{c-1}, \ldots, \frac{\eta}{c-1})$ is too high since the true minimum achievable risk is lower than this bound. In other words, there exists a probability estimator which attains a risk lower than our bound. This non-optimality is the cost we incur as a result of requiring a simple, easily computable bound depending on only on the noise rate. Importantly, Corollary D.1 gives us a rough way to quantify this non-optimality, using the

difference between the upper and lower entropy limits

$$\left[ \mathcal{H}\left(1 - \eta, \frac{\eta}{c-1}, \frac{\eta}{c-1}, \ldots, \frac{\eta}{c-1}\right), \mathcal{H}(1 - \eta, \eta, 0, 0, \ldots, 0) \right] \tag{14}$$

When this difference is large, one can construct two types of label noise with the same rate $\eta$, such that the difference in the minimum achievable risks between these noise types is significant. Conversely, when this gap is small, the minimum achievable risk for any type of label noise at a fixed rate $\eta$ is similar. This is a desirable property and suggests that simply setting our bound to our noise-bound is probably suitable regardless of the specifics of the noising process.

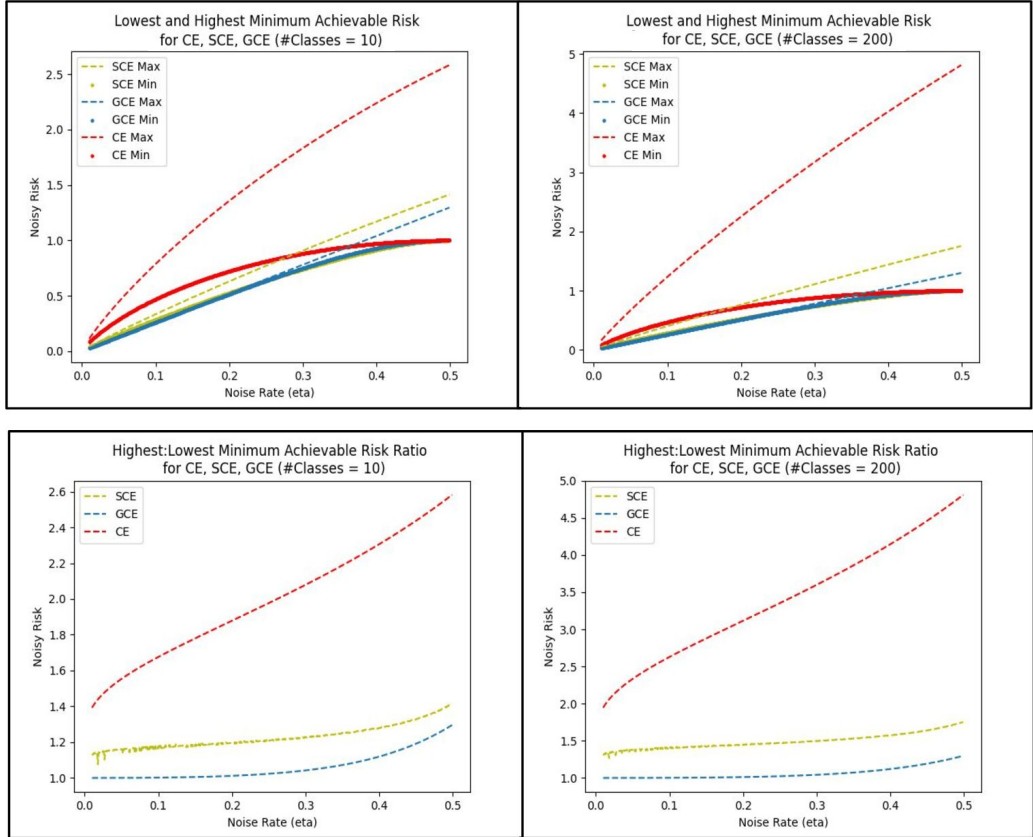

Figure 3: On the top row, we plot the upper and lower limits of the interval given in Equation 14 for $\eta \in (0, 0.5]$ in the proof of Corollary D.1 for the CE (red), SCE (yellow) and GCE (blue) losses for 10 classes (left) and 200 classes (right). On the bottom row, we plot a ratio of these upper and lower limits instead. We observe that the difference between these upper and lower limits is far greater for CE than the other losses. This is more pronounced for more classes.

On the top row of Figure 3, we give a plot of the upper and lower limits of Equation 14 for $\eta \in (0, 0.5]$ for $c = 10$ (left) and $c = 200$ (right) for GCE, SCE and CE. The upper limit is given by a dotted line, while the lower limit is given by a filled line in the same colour. Each loss is scaled so they may be more easily compared. Similarly, in the row below, we plot the ratios of the upper and lower limits of Equation 14 for each loss. These graphs show that the difference between the upper and lower limits is much greater for CE than for SCE and GCE. This difference is more pronounced when the number of classes is greater. **The result is that on asymmetric noise, our noise-bound (Definition 4.5) will generally be less suitable when used in conjunction with CE than when used with GCE or SCE.**

### D.2.1 Noise Model Ablation

In Section D.2, we explained how noise bounding is likely to be less effective for asymmetric label noise models than symmetric noise. Moreover, it is likely to be especially ineffective for the cross-entropy loss since the Jensen-Shannon entropy varies greatly even for distributions with the same noise rate (Figure 3). In this section, we validate this hypothesis experimentally, showing the superior performance of noise-bounding with SCE compared to CE in asymmetric noise settings. Findings are summarized in Figure 4.

**Experiment Setup:** Lemma 4.3 tells us that, for a fixed noise rate, the highest and lowest entropies occur with symmetric and pairwise label noise models, respectively. We fix a noise rate of $\eta = 0.4$ and interpolate between these noise models. As we interpolate from symmetric to pairwise noise, we evaluate the performance of noise-bounding for the cross-entropy (CE) and Symmetric Cross-Entropy (SCE) loss functions versus vanilla CE and SCE. Specifically, we use the given noise model to noise the labels of the training set, we then train using each loss for a set number of epochs, and record the final clean test accuracy obtained. The results are shown in Figure 4, SCE on the left and CE on the right. The unbounded variant of each loss is shown in blue and the noise-bounded variant is in red. For each graph, symmetric label noise is represented at the leftmost of the figure and pairwise label noise at the furthest right. Thus, the label noise model becomes more asymmetric as we move right of each graph in Figure 4. As the label noise become more asymmetric, performance deteriorates for all loss functions. In contrast, for the SCE loss, noise-bounding grants improved robustness over standard, unbounded, SCE up to (but not including) pairwise asymmetric label noise. In contrast, for CE, as anticipated, performance deteriorates dramatically as the noise becomes less symmetric.

**Experiment Details** We use a noised variant of the EMNIST dataset wherein we noise the training set using each noise model while keeping the test set clean for evaluation. We train for 80 epochs. Our experiment requires us to interpolate between symmetric and pairwise label noise. Symmetric label noise allows a label to transition to any other label with equal probability. For pairwise noise, transitions occur between pairs of labels. We interpolate between these noise types by allowing transitions within sets of labels of size $m$, where $m$ satisfies $2 < m < c$ ($c$ being the total number of labels, in our case $c = 47$).

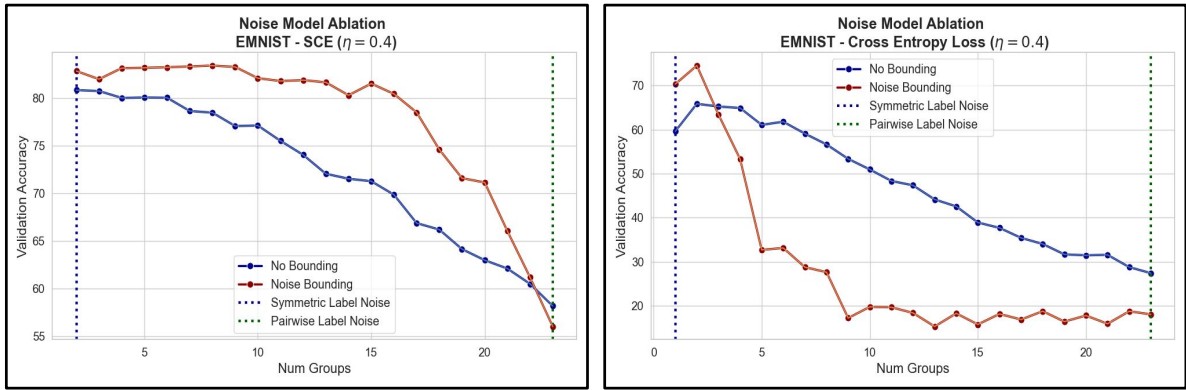

Figure 4: **Impact of Noise Model on Performance with a Fixed Noise Rate ($\eta = 0.4$):** We fix the noise rate at $\eta = 0.4$ and examine the effectiveness of our approach across different noise models at this rate. We interpolate between symmetric label noise (on the left-hand side of each graph) and pairwise label noise (on the right-hand side), for the cross-entropy and SCE loss functions. As the noise model becomes more asymmetric, the performance of our approach deteriorates. This deterioration is expected since the noise bound is calculated based on an idealized assumption that label noise is symmetric. The magnitude of this deterioration is initially small for the SCE loss; noise-bounding still provides a benefit when label noise is slightly asymmetric compared to not employing a bound. However, when the label noise is completely pairwise, noise-bounding with the SCE no longer provides a benefit. More starkly, the performance of the noise-bounded cross-entropy becomes significantly reduced as the noise model becomes more asymmetric.

### D.3  Noise Model Plots

In Lemma 2.2 we showed that the SCE, GCE and FCE losses are generalised forward-correction losses and derived the corresponding functions $f$. (In fact we derived $f^{-1}$ as this turned out to be easier.) As discussed, these functions can be interpreted as noise models; $f(\boldsymbol{p}(y|x)) \approx \tilde{\boldsymbol{p}}(\tilde{y}|x)$. In section we provide some plots of these noise models.

**Properness**  While Definition 2.1 does not require the so-called base loss to be proper, Lemma 2.2 shows that GCE and SCE can be obtained by applying a non-linear correction to a proper loss. The defining characteristic of a proper loss is that the expected loss is minimised by setting $\boldsymbol{p} = \boldsymbol{q}$. Therefore,

$$H_{L_f}(\tilde{\boldsymbol{p}}, \boldsymbol{q}) := \tilde{\boldsymbol{p}} \cdot L_f(\boldsymbol{q}) = \tilde{\boldsymbol{p}} \cdot L(f(\boldsymbol{q})).$$

is minimised by setting $\tilde{\boldsymbol{p}} = f(\boldsymbol{q}) \iff \boldsymbol{q} = f^{-1}(\tilde{\boldsymbol{p}})$. We make this point because plotting $f^{-1}$ as a function of $\boldsymbol{p}$ (which we do below) is the same as plotting $\arg\min_q H(\boldsymbol{p}, \boldsymbol{q})$ - this allows us to include the MAE loss on this plot even though it isn't a generalised forward-correction loss.

**Plots**  In Figure 5, we present plots of $f^{-1}$ for the SCE, GCE and FCE loss functions in the binary setting. The $x-$axis gives probability of a noisy label being equal to one $\tilde{p}(\tilde{y} = 1 \mid x)$. On the $y-$axis we plot $p(y = 1 \mid x)$ where $\boldsymbol{p} = f^{-1}(\tilde{\boldsymbol{p}})$. For proper losses, $f = id$, reflecting the fact that they contain encode no noise model. The graphs for GCE and SCE are remarkably similar. Their graphs portray a noise model where labels noise occurs more frequently at points where $\boldsymbol{p}$ contains higher intrinsic uncertainty. Conversely no label noise occurs at anchor points at all. FCE requires a noise model in order to be fully specified; we assume symmetric label noise at $\eta = 0.4$. Varying $\eta$ will change the steepness of the respective $f^{-1}$. Finally, we plot MAE The graphs of SCE and GCE lie between those of MAE and CE. By varying the parameters of these losses, we can interpolate between them.

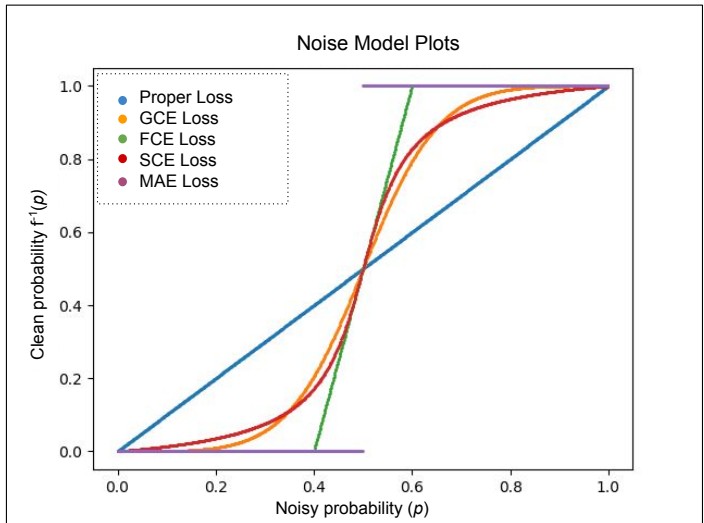

Figure 5: Plot of $f^{-1}(p)$ for SCE ($A = 8$), GCE ($a = 0.7$), FCE ($\eta = 0.4$), CE and MAE in the binary case. We have the true probability $p$ on the x-axis and the choice of $q$, which minimises the expected loss on the $y$-axis.

### D.4 Explicit Bounds

Using Lemma 2.2 we can produce the noise-bounds (Definition 4.5) for GCE and SCE. The bound for GCE is given below.

$$B_{GCE}(\eta, c) := \frac{(1-\eta)}{a} \left( 1 - \left( \frac{(1-\eta)^{\frac{1}{1-a}}}{(1-\eta)^{\frac{1}{1-a}} + (c-1)(\frac{\eta}{c-1})^{\frac{1}{1-a}}} \right)^a \right) +$$

$$\frac{\eta}{a} \left( 1 - \left( \frac{\frac{\eta}{c-1}^{\frac{1}{1-a}}}{(1-\eta)^{\frac{1}{1-a}} + (c-1)(\frac{\eta}{c-1})^{\frac{1}{1-a}}} \right)^a \right).$$

The noise-bound for SCE is

$$B_{SCE}(\eta, c) := (1-\eta) \left( -log \left( \frac{1-\eta}{\lambda - A(1-\eta)} \right) + A \left( 1 - \frac{1-\eta}{\lambda - A(1-\eta)} \right) \right)$$

$$+\eta \left( -log \left( \frac{\eta}{\lambda(c-1) - A\eta} \right) + A \left( 1 - \frac{\eta}{\lambda(c-1) - A\eta} \right) \right).$$

Recollect that $\lambda$ is chosen so that the resulting distribution normalises: $\frac{1-\eta}{\lambda - A(1-\eta)} + \frac{\eta(c-1)}{\lambda(c-1) - A\eta} = 1$ and may be computed numerically or by solving the resulting quadratic.

## E   Further Experiments

### E.1   Experiment Details

The number of training epochs was the same for each loss. For MNIST, FashionMNIST, TinyImageNet and Animals10N, we used 100 epochs; for all other datasets, we used 120 epochs. Each experiment in Tables 1 and 2 (Section 6) was run three times, and the mean and unbiased estimate of the standard deviation is given. We used a ResNet18 architecture for all experiments except TinyImageNet and Animals10N, where a ResNet34 was used. Each experiment is carried out on a single GeForce GTX Titan X. We used a batch size of 300 in all experiments except TinyImageNet and Animals10N, where this is reduced to 200. A learning rate of 0.0001 was used for all losses except MAE (lr = 0.001) and ELR where we used their recommended learning rate of 0.01. We use a learning rate scheduler which scales our learning rate by 0.6 at epoch 60. Our implementation of the *Truncated Loss* comes from the official github implementation of GCE. Likewise, we use the official codebase for our implementation of ELR. Other losses are re-implementations based on details given in the respective papers. Our SCE loss used the recommended hyperparameter of $A = 8$. Our GCE loss used $a = 0.4$. FCE requires one to define a noise model. In each case, we assume noise is symmetric at the relevant rate. For Animals10N, this rate is set to 11%, which is the estimated noise rate.

#### E.1.1   CE with Prior

One of the losses used in our experiments is cross-entropy with a 'prior' term (CEP). We give an explanation of the motivation for this additional loss term and details of how it's implemented.

In Section 4.2 we assumed that the un-noised distribution $p(x, y)$ is separable (i.e. for each $x$, $p(y = k \mid x) = 1, p(i \neq k \mid x) = 0$) for some $k \in \mathcal{Y}$. Thus, in the case of symmetric noise with a known noise rate $\eta$, the noisy label distribution $\widetilde{p}(\widetilde{y}|x)$ is of the form for each $x$:

$$\widetilde{p}(\widetilde{y} \mid x) = \left( \frac{\eta}{c-1}, \frac{\eta}{c-1}, \dots, \underbrace{1-\eta}_{k^{\text{th}} \text{ position}}, \dots, \frac{\eta}{c-1} \right) \tag{15}$$

We argue, therefore, that it is reasonable to introduce a term to penalise our model when its outputs deviate from this distribution. This is achieved through a regularisation term which measures the KL-divergence between our model probabilities and the desired distribution (Equation 15). Let $\boldsymbol{p_\eta} := (p_1, p_2, \dots, p_c) :=$

$(1 - \eta, \frac{\eta}{c-1}, \ldots, \frac{\eta}{c-1})$ and let $q_1, q_2, \ldots, q_c$ denote the probabilities output by our model. We sort the $q_i$ into descending order (which we denote as $q_{\sigma(i)}$) and define our prior term as:

$$L_{prior}(\boldsymbol{q}, \boldsymbol{p}_\eta) := -\sum_{i=1}^{c} p_i \log(q_{\sigma(i)}) \qquad (16)$$

Thus, overall we have $L_{CEP}(\boldsymbol{q}, i) := L_{CE}(\boldsymbol{q}, i) + L_{prior}(\boldsymbol{q}, \boldsymbol{p}_\eta)$. Tables 1 and 2 in Section 6 show that this additional term generally results in additional improvement over using the noise-bound alone. This prior acts as a method of feasible set reduction: There are many different probability estimators which achieve a training error equal to our noise-bound. Therefore, by introducing a prior term (Equation 16) we can further restrict the set of admissible models.

## E.2 Varying The Bound

We explore treating the bound '$B$' as a hyperparameter to assess the proximity of the noise-bound to optimality. This consists of doing a grid search in a small vicinity of the noise-bound for each loss function and recording how this impacts clean test performance. Tables 7, 8 include the result of these experiments, indicated with a star (e.g. CE+B*) together with the results of the other loss functions. When varying the bound from the noise-bound doesn't yield an improvement, the starred and unstarred accuracy values are the same. In slightly over half of our experiments, we find that we may achieve an improvement by perturbing the bound. This improvement is generally minor. Our assumption that the underlying clean dataset is separable means one should be able to improve performance by raising the bound to account for the additional randomness in the label distributions. Generally, we find this to be so. An exception to this pattern are the non-uniform and asymmetric datasets. In these cases, one typically benefits from marginally *lowering* the bound. This observation is consistent with our expectation; the noise-bound is a 'worst-case' entropy, attained only by uniform symmetric label noise. For other noise models the noise-bound will be higher than strictly necessary to prevent overfitting and may benefit from being slightly decreased. The values of the optimal bounds may be found in a table in Appendix E.

|     | MNIST | | Fashion | | EMNIST | | CIFAR10 | | CIFAR100 | | ACIFAR100 | | NU-EMNIST |
|-----|-------|------|---------|-------|--------|------|---------|------|----------|-------|-----------|-------|-----------|
|     | 0.4   | 0.6  | 0.2     | 0.4   | 0.2    | 0.4  | 0.2     | 0.4  | 0.2      | 0.4   | 0.2       | 0.4   | 0.6       |
| FCE | -0.05 | -0.05| 0.0     | -0.05 | 0.05   | 0.05 | 0.05    | 0.1  | 0.03     | 0.0   | -0.1      | -0.35 | -0.2      |
| GCE | 0.0   | 0.0  | 0.05    | 0.0   | 0.03   | 0.05 | 0.05    | 0.05 | 0.05     | 0.0   | 0.05      | 0.05  | 0.0       |
| SCE | 0.0   | 0.05 | 0.0     | 0.2   | 0.2    | 0.1  | 0.0     | 0.2  | 0.2      | 0.2   | 0.0       | 0.0   | 0.0       |
| CEB | 0.0   | 0.0  | 0.0     | 0.0   | 0.05   | 0.05 | 0.05    | 0.05 | 0.1      | 0.1   | 0.2       | 0.0   | -0.6      |
| CEP | -0.15 | -0.15| -0.15   | -0.15 | 0.0    | 0.02 | 0.0     | 0.0  | -0.08    | -0.08 | -0.08     | -0.08 | -0.1      |

Table 6: table giving the offset of the 'optimal' bound from the noise-bound. Here a negative (blue) number means that the bound is greater than the noise-bound. Positive (red) means the optimal bound is lower. Grey means that the optimal bound is zero, i.e. no offset.

### E.2.1 Optimal Bounds

In our experiment tables in Section E.2, we give results using our noise-bounds. We additionally give results where the bound is treated as a hyperparameter. We do not search over the entire space; rather, we do a grid search near the noise-bound. For MNIST, FashionMNIST, EMNIST, CIFAR10 and CIFAR100, we search over $\{-0.2, -0.15, -0.1, \ldots, 0.15, 0.2\}$ where e.g. 0.2 means that we add 0.2 onto our noise-bound $(B(\eta, c) \mapsto B(\eta, c) + 0.2)$. For Asymmetric CIFAR100 (ACIFAR100) and Non-uniform EMNIST (NU-EMNIST), this range is broadened to $\{-0.6, -0.55, \ldots 0.55, 0.6\}$. The bounds which give the best results are given in Table 6. When the optimal bound is higher than the noise-bound, this is highlighted in blue. Otherwise, the cell is indicated in red. In our original table, we have columns for Top1 and Top5 accuracy which often have slightly different optimal bounds. For brevity, we combine these by taking a mean of these values.

| Losses | MNIST 0.4 | MNIST 0.6 | FashionMNIST 0.2 | FashionMNIST 0.4 | EMNIST 0.2 Top 1 | EMNIST 0.2 Top 5 | EMNIST 0.4 Top 1 | EMNIST 0.4 Top 5 | CIFAR10 0.2 | CIFAR10 0.4 |
|---|---|---|---|---|---|---|---|---|---|---|
| MSE | $93.3_{\pm0.47}$ | $85.8_{\pm0.95}$ | $84.8_{\pm0.22}$ | $80.6_{\pm0.84}$ | $82.9_{\pm0.29}$ | $98.1_{\pm0.04}$ | $80.2_{\pm0.19}$ | $97.1_{\pm0.07}$ | $78.7_{\pm1.51}$ | $56.4_{\pm0.11}$ |
| MAE | $97.9_{\pm0.08}$ | $96.4_{\pm0.08}$ | $83.2_{\pm0.10}$ | $82.2_{\pm0.37}$ | $49.8_{\pm2.83}$ | $52.2_{\pm0.10}$ | $50.4_{\pm1.14}$ | $51.4_{\pm0.96}$ | $88.6_{\pm1.34}$ | $78.9_{\pm5.95}$ |
| NCE | $97.8_{\pm0.06}$ | $96.0_{\pm0.25}$ | $87.7_{\pm0.26}$ | $\mathbf{86.3}_{\pm0.14}$ | $84.5_{\pm0.25}$ | $97.9_{\pm0.05}$ | $82.6_{\pm0.81}$ | $96.7_{\pm0.03}$ | $\mathbf{89.3}_{\pm0.40}$ | $\mathbf{86.0}_{\pm0.81}$ |
| MixUp | $95.8_{\pm1.24}$ | $86.8_{\pm0.85}$ | $86.9_{\pm0.10}$ | $82.3_{\pm0.54}$ | $84.6_{\pm0.08}$ | $98.1_{\pm0.04}$ | $81.6_{\pm0.48}$ | $97.1_{\pm0.04}$ | $86.0_{\pm0.46}$ | $77.9_{\pm0.49}$ |
| Spher. | $95.0_{\pm0.41}$ | $88.1_{\pm0.82}$ | $87.2_{\pm0.04}$ | $84.1_{\pm0.75}$ | $84.6_{\pm0.12}$ | $98.3_{\pm0.05}$ | $83.2_{\pm0.29}$ | $98.1_{\pm0.58}$ | $86.6_{\pm0.01}$ | $72.1_{\pm0.80}$ |
| Boot. | $86.6_{\pm0.56}$ | $71.2_{\pm1.17}$ | $82.0_{\pm0.61}$ | $73.4_{\pm1.06}$ | $80.5_{\pm0.24}$ | $96.7_{\pm0.06}$ | $77.3_{\pm0.98}$ | $95.0_{\pm0.25}$ | $77.0_{\pm1.57}$ | $58.2_{\pm2.99}$ |
| Trunc. | $97.1_{\pm0.12}$ | $94.2_{\pm0.39}$ | $87.8_{\pm0.29}$ | $85.3_{\pm0.77}$ | $84.1_{\pm0.53}$ | $97.4_{\pm1.03}$ | $83.1_{\pm0.55}$ | $97.2_{\pm1.00}$ | $88.3_{\pm0.56}$ | $84.2_{\pm0.69}$ |
| CL | $82.7_{\pm0.57}$ | $67.5_{\pm1.83}$ | $81.2_{\pm0.34}$ | $73.1_{\pm0.66}$ | $79.6_{\pm0.17}$ | $96.4_{\pm0.05}$ | $75.1_{\pm0.67}$ | $94.2_{\pm0.24}$ | $76.0_{\pm2.16}$ | $59.4_{\pm4.20}$ |
| ELR | $98.1_{\pm0.04}$ | $97.8_{\pm0.07}$ | $85.3_{\pm0.23}$ | $83.4_{\pm0.02}$ | $81.8_{\pm0.26}$ | $97.5_{\pm0.21}$ | $76.6_{\pm0.10}$ | $96.5_{\pm0.11}$ | $88.1_{\pm0.82}$ | $85.7_{\pm0.06}$ |
| FCE. | $95.4_{\pm0.25}$ | $92.3_{\pm0.13}$ | $83.6_{\pm0.11}$ | $79.9_{\pm0.78}$ | $83.1_{\pm0.12}$ | $98.4_{\pm0.20}$ | $80.6_{\pm0.12}$ | $98.0_{\pm0.03}$ | $84.7_{\pm0.40}$ | $75.1_{\pm0.04}$ |
| FCE+B | $95.7_{\pm0.18}$ | $92.7_{\pm0.74}$ | $84.8_{\pm0.26}$ | $81.7_{\pm0.27}$ | $83.4_{\pm0.09}$ | $98.5_{\pm0.03}$ | $81.6_{\pm0.51}$ | $\mathbf{98.1}_{\pm0.15}$ | $86.7_{\pm0.21}$ | $82.2_{\pm0.06}$ |
| FCE+B* | $96.7_{\pm0.17}$ | $94.3_{\pm0.50}$ | $84.8_{\pm0.26}$ | $83.3_{\pm0.22}$ | $84.4_{\pm0.06}$ | $98.6_{\pm0.13}$ | $83.1_{\pm0.42}$ | $98.1_{\pm0.10}$ | $87.2_{\pm0.20}$ | $82.2_{\pm0.06}$ |
| GCE | $94.4_{\pm0.36}$ | $83.8_{\pm1.14}$ | $86.4_{\pm0.24}$ | $81.6_{\pm0.37}$ | $84.3_{\pm0.13}$ | $98.4_{\pm0.08}$ | $82.7_{\pm0.07}$ | $97.9_{\pm0.02}$ | $81.1_{\pm0.72}$ | $60.0_{\pm1.31}$ |
| GCE+B | $96.6_{\pm0.22}$ | $94.0_{\pm0.13}$ | $86.5_{\pm0.56}$ | $85.5_{\pm0.13}$ | $84.1_{\pm0.29}$ | $98.4_{\pm0.04}$ | $82.8_{\pm0.28}$ | $98.0_{\pm0.06}$ | $86.1_{\pm0.22}$ | $79.0_{\pm1.17}$ |
| GCE+B* | $96.6_{\pm0.22}$ | $94.0_{\pm0.13}$ | $87.0_{\pm0.04}$ | $85.5_{\pm0.13}$ | $84.3_{\pm0.09}$ | $98.4_{\pm0.06}$ | $83.6_{\pm0.25}$ | $\mathbf{98.2}_{\pm0.03}$ | $86.7_{\pm0.07}$ | $80.2_{\pm0.83}$ |
| SCE | $89.5_{\pm5.29}$ | $70.2_{\pm0.69}$ | $82.7_{\pm0.64}$ | $74.4_{\pm0.37}$ | $82.1_{\pm0.33}$ | $96.8_{\pm0.10}$ | $79.6_{\pm0.61}$ | $95.4_{\pm0.15}$ | $78.2_{\pm0.42}$ | $59.0_{\pm4.43}$ |
| SCE+B | $97.0_{\pm0.16}$ | $93.4_{\pm0.29}$ | $87.5_{\pm0.22}$ | $85.2_{\pm0.98}$ | $83.5_{\pm0.29}$ | $97.3_{\pm0.14}$ | $81.8_{\pm0.52}$ | $96.4_{\pm0.20}$ | $88.9_{\pm0.44}$ | $84.7_{\pm0.37}$ |
| SCE+B* | $97.0_{\pm0.16}$ | $93.7_{\pm0.52}$ | $87.5_{\pm0.22}$ | $85.8_{\pm0.67}$ | $83.6_{\pm0.03}$ | $97.4_{\pm0.02}$ | $81.8_{\pm0.52}$ | $96.5_{\pm0.26}$ | $88.9_{\pm0.44}$ | $84.9_{\pm0.20}$ |
| CE | $80.8_{\pm2.31}$ | $67.3_{\pm0.80}$ | $80.9_{\pm1.11}$ | $72.1_{\pm2.16}$ | $79.9_{\pm0.28}$ | $96.4_{\pm0.08}$ | $75.6_{\pm0.20}$ | $94.2_{\pm0.24}$ | $76.9_{\pm1.22}$ | $59.9_{\pm2.15}$ |
| CE+B | $96.2_{\pm0.32}$ | $93.0_{\pm0.09}$ | $87.9_{\pm0.10}$ | $84.7_{\pm0.37}$ | $80.8_{\pm0.08}$ | $97.0_{\pm0.04}$ | $78.9_{\pm0.12}$ | $96.1_{\pm0.26}$ | $84.5_{\pm0.73}$ | $76.0_{\pm1.13}$ |
| CE+B* | $96.2_{\pm0.32}$ | $93.0_{\pm0.09}$ | $87.9_{\pm0.10}$ | $84.7_{\pm0.37}$ | $81.5_{\pm0.11}$ | $97.3_{\pm0.02}$ | $79.0_{\pm0.09}$ | $96.2_{\pm0.01}$ | $84.8_{\pm0.55}$ | $78.6_{\pm1.28}$ |
| CEP | $97.5_{\pm0.08}$ | $92.1_{\pm0.44}$ | $87.8_{\pm0.12}$ | $84.8_{\pm0.23}$ | $85.5_{\pm0.10}$ | $98.1_{\pm0.07}$ | $84.3_{\pm0.22}$ | $97.6_{\pm0.14}$ | $84.2_{\pm0.51}$ | $58.2_{\pm2.94}$ |
| CEP+B | $95.6_{\pm0.32}$ | $85.5_{\pm0.77}$ | $88.1_{\pm0.31}$ | $84.2_{\pm0.33}$ | $\mathbf{85.8}_{\pm0.12}$ | $98.3_{\pm0.02}$ | $\mathbf{84.8}_{\pm0.10}$ | $98.0_{\pm0.04}$ | $88.5_{\pm0.32}$ | $85.1_{\pm0.20}$ |
| CEP+B* | $\mathbf{98.5}_{\pm0.05}$ | $\mathbf{97.9}_{\pm0.11}$ | $\mathbf{88.4}_{\pm0.04}$ | $\mathbf{87.2}_{\pm0.21}$ | $85.8_{\pm0.12}$ | $98.3_{\pm0.02}$ | $84.8_{\pm0.10}$ | $98.0_{\pm0.16}$ | $88.5_{\pm0.32}$ | $85.1_{\pm0.20}$ |

Table 7: Test accuracies obtained by using different losses on the noisy MNIST/ FashionM-NIST/EMNIST/CIFAR10 datasets. Losses implementing the noise-bound shaded in blue. When using this bound provides benefit, the corresponding value is boxed. Overall top values in **bold**.

| Losses | CIFAR100 0.2 Top1 | CIFAR100 0.2 Top5 | CIFAR100 0.4 Top1 | CIFAR100 0.4 Top5 | ASYM-CIFAR100 0.2 Top1 | ASYM-CIFAR100 0.2 Top5 | ASYM-CIFAR100 0.4 Top1 | ASYM-CIFAR100 0.4 Top5 | Non-Uniform-EMNIST 0.6 Top 1 | Non-Uniform-EMNIST 0.6 Top 5 |
|---|---|---|---|---|---|---|---|---|---|---|
| MSE | $57.2_{\pm0.93}$ | $78.6_{\pm0.25}$ | $40.6_{\pm0.38}$ | $63.0_{\pm0.24}$ | $56.3_{\pm0.11}$ | $82.6_{\pm0.22}$ | $40.7_{\pm0.12}$ | $74.4_{\pm0.25}$ | $44.7_{\pm2.66}$ | $86.7_{\pm3.10}$ |
| MAE | $10.0_{\pm0.11}$ | $13.8_{\pm0.28}$ | $7.6_{\pm1.89}$ | $11.6_{\pm1.25}$ | $7.1_{\pm6.02}$ | $11.1_{\pm6.6}$ | $11.1_{\pm5.43}$ | $25.1_{\pm5.76}$ | $9.8_{\pm1.74}$ | $23.1_{\pm1.80}$ |
| NCE | $38.7_{\pm3.13}$ | $51.8_{\pm3.77}$ | $19.1_{\pm0.20}$ | $28.8_{\pm0.15}$ | $16.3_{\pm1.24}$ | $25.4_{\pm1.80}$ | $21.8_{\pm1.24}$ | $37.2_{\pm1.80}$ | $18.0_{\pm1.17}$ | $38.8_{\pm1.93}$ |
| MixUp | $59.6_{\pm0.31}$ | $81.5_{\pm0.39}$ | $51.3_{\pm8.63}$ | $75.8_{\pm8.09}$ | $61.2_{\pm0.88}$ | $86.0_{\pm1.12}$ | $47.2_{\pm0.60}$ | $81.3_{\pm0.23}$ | $\mathbf{52.4}_{\pm0.80}$ | $95.5_{\pm0.08}$ |
| Spher. | $57.7_{\pm0.18}$ | $82.9_{\pm0.54}$ | $48.8_{\pm8.51}$ | $74.3_{\pm0.73}$ | $54.2_{\pm0.32}$ | $81.2_{\pm0.29}$ | $39.2_{\pm0.31}$ | $72.1_{\pm0.15}$ | $41.9_{\pm0.10}$ | $94.4_{\pm0.04}$ |
| Boot. | $54.0_{\pm0.37}$ | $76.4_{\pm0.39}$ | $37.7_{\pm0.89}$ | $60.9_{\pm1.52}$ | $56.0_{\pm0.34}$ | $83.8_{\pm0.03}$ | $43.2_{\pm0.35}$ | $78.3_{\pm0.20}$ | $49.1_{\pm0.29}$ | $95.3_{\pm0.42}$ |
| Trunc. | $58.1_{\pm0.36}$ | $82.7_{\pm0.37}$ | $50.9_{\pm1.17}$ | $77.2_{\pm0.59}$ | $56.3_{\pm0.62}$ | $82.3_{\pm0.61}$ | $45.2_{\pm0.81}$ | $75.6_{\pm0.29}$ | $23.7_{\pm0.98}$ | $40.1_{\pm1.24}$ |
| CL | $53.0_{\pm0.21}$ | $76.3_{\pm0.19}$ | $36.3_{\pm0.77}$ | $60.1_{\pm0.66}$ | $55.3_{\pm0.48}$ | $83.5_{\pm0.28}$ | $42.4_{\pm0.45}$ | $78.1_{\pm0.14}$ | $48.2_{\pm0.45}$ | $95.0_{\pm0.04}$ |
| ELR | $10.4_{\pm0.24}$ | $31.7_{\pm0.44}$ | $10.0_{\pm0.64}$ | $30.1_{\pm0.88}$ | $18.8_{\pm0.21}$ | $32.7_{\pm0.53}$ | $10.3_{\pm0.39}$ | $30.8_{\pm0.35}$ | $40.3_{\pm0.39}$ | $93.0_{\pm0.24}$ |
| FCE | $56.9_{\pm0.58}$ | $79.2_{\pm0.14}$ | $43.7_{\pm0.15}$ | $66.2_{\pm0.19}$ | $55.3_{\pm0.54}$ | $83.5_{\pm0.24}$ | $41.4_{\pm0.55}$ | $77.3_{\pm0.75}$ | $39.0_{\pm0.05}$ | $67.8_{\pm0.47}$ |
| FCE+B | $56.1_{\pm2.22}$ | $81.8_{\pm1.37}$ | $50.2_{\pm0.02}$ | $77.2_{\pm0.19}$ | $54.2_{\pm0.44}$ | $83.3_{\pm0.43}$ | $43.8_{\pm0.02}$ | $77.5_{\pm0.13}$ | $40.0_{\pm0.35}$ | $73.2_{\pm0.08}$ |
| FCE+B* | $56.1_{\pm2.22}$ | $82.2_{\pm0.39}$ | $50.2_{\pm0.02}$ | $77.2_{\pm0.19}$ | $54.2_{\pm0.44}$ | $83.4_{\pm0.24}$ | $45.1_{\pm0.37}$ | $79.9_{\pm0.24}$ | $43.1_{\pm0.40}$ | $79.4_{\pm0.12}$ |
| GCE | $60.0_{\pm0.13}$ | $82.6_{\pm0.63}$ | $44.9_{\pm0.07}$ | $67.2_{\pm0.34}$ | $53.8_{\pm0.55}$ | $81.6_{\pm0.14}$ | $39.4_{\pm0.44}$ | $74.0_{\pm0.36}$ | $44.8_{\pm0.62}$ | $91.2_{\pm0.70}$ |
| GCE+B | $59.4_{\pm0.02}$ | $83.5_{\pm0.24}$ | $50.3_{\pm0.11}$ | $75.3_{\pm0.64}$ | $55.4_{\pm0.55}$ | $83.0_{\pm0.35}$ | $46.5_{\pm1.44}$ | $77.7_{\pm0.35}$ | $47.1_{\pm0.20}$ | $93.5_{\pm0.43}$ |
| GCE+B* | $61.0_{\pm1.33}$ | $83.9_{\pm0.74}$ | $50.3_{\pm0.11}$ | $75.3_{\pm0.64}$ | $56.6_{\pm0.10}$ | $83.8_{\pm0.88}$ | $47.7_{\pm0.35}$ | $77.9_{\pm0.03}$ | $47.1_{\pm0.20}$ | $93.5_{\pm0.43}$ |
| SCE | $55.9_{\pm0.53}$ | $76.5_{\pm0.15}$ | $38.7_{\pm0.60}$ | $60.9_{\pm0.41}$ | $57.5_{\pm0.19}$ | $83.7_{\pm0.17}$ | $43.3_{\pm0.87}$ | $77.5_{\pm0.75}$ | $47.2_{\pm0.33}$ | $92.5_{\pm0.01}$ |
| SCE+B | $55.5_{\pm0.90}$ | $77.4_{\pm0.84}$ | $47.1_{\pm1.32}$ | $69.2_{\pm1.18}$ | $57.9_{\pm0.83}$ | $83.7_{\pm0.41}$ | $50.0_{\pm1.62}$ | $80.4_{\pm0.65}$ | $47.9_{\pm0.80}$ | $93.8_{\pm0.05}$ |
| SCE+B* | $56.6_{\pm1.07}$ | $78.5_{\pm0.88}$ | $47.3_{\pm1.16}$ | $69.6_{\pm0.90}$ | $57.9_{\pm0.83}$ | $83.7_{\pm0.41}$ | $50.0_{\pm1.62}$ | $80.4_{\pm0.65}$ | $47.9_{\pm0.80}$ | $93.8_{\pm0.05}$ |
| CE | $52.3_{\pm1.35}$ | $75.6_{\pm0.93}$ | $35.3_{\pm1.14}$ | $59.3_{\pm0.81}$ | $54.9_{\pm0.12}$ | $83.3_{\pm0.25}$ | $42.4_{\pm0.16}$ | $78.9_{\pm0.56}$ | $48.6_{\pm0.11}$ | $95.3_{\pm0.10}$ |
| CE+B | $50.9_{\pm1.01}$ | $76.5_{\pm0.86}$ | $39.9_{\pm1.02}$ | $65.8_{\pm1.19}$ | $52.9_{\pm1.86}$ | $83.2_{\pm0.88}$ | $34.7_{\pm2.51}$ | $73.4_{\pm1.50}$ | $45.5_{\pm5.11}$ | $93.0_{\pm0.16}$ |
| CE+B* | $50.9_{\pm1.01}$ | $78.2_{\pm1.16}$ | $39.9_{\pm1.02}$ | $68.1_{\pm0.63}$ | $53.3_{\pm0.89}$ | $83.2_{\pm0.88}$ | $45.9_{\pm0.40}$ | $79.7_{\pm0.29}$ | $50.2_{\pm0.35}$ | $\mathbf{95.9}_{\pm0.14}$ |
| CEP | $58.8_{\pm0.87}$ | $78.6_{\pm0.38}$ | $43.5_{\pm0.24}$ | $65.1_{\pm1.27}$ | $59.4_{\pm0.08}$ | $82.2_{\pm0.03}$ | $46.5_{\pm0.17}$ | $76.4_{\pm0.25}$ | $48.2_{\pm0.05}$ | $95.4_{\pm0.07}$ |
| CEP+B | $\mathbf{62.3}_{\pm0.87}$ | $\mathbf{85.1}_{\pm0.46}$ | $54.3_{\pm0.86}$ | $79.2_{\pm0.93}$ | $\mathbf{63.0}_{\pm0.92}$ | $\mathbf{87.5}_{\pm0.32}$ | $53.0_{\pm0.28}$ | $82.8_{\pm0.13}$ | $45.0_{\pm0.48}$ | $95.0_{\pm0.08}$ |
| CEP+B* | $62.9_{\pm0.79}$ | $85.1_{\pm0.46}$ | $\mathbf{55.3}_{\pm0.37}$ | $\mathbf{79.8}_{\pm0.08}$ | $\mathbf{63.0}_{\pm0.14}$ | $87.5_{\pm0.32}$ | $\mathbf{55.6}_{\pm0.66}$ | $\mathbf{83.8}_{\pm0.11}$ | $47.7_{\pm0.19}$ | $95.9_{\pm0.23}$ |

Table 8: Test accuracies for different losses on the noisy CIFAR100/Asym-CIFAR100/Non-Uniform EMNIST datasets. Losses implementing the noise-bound shaded in blue. When using this bound provides benefit, the corresponding value is boxed. Overall top values in **bold**.

| Losses | TinyImageNet (0.2) | | TinyImageNet (0.4) | | Animals |
|---|---|---|---|---|---|
| | Top 1 | Top 5 | Top 1 | Top 5 | |
| L2 (MSE) | 42.91 | 67.02 | 29.42 | 53.13 | 80.97 |
| MAE | 3.86 | 5.58 | 3.94 | 5.54 | 54.67 |
| NCE-MAE | 7.63 | 10.24 | 6.29 | 10.70 | 80.85 |
| Mix-Up | 47.13 | 70.08 | 31.05 | 58.96 | 83.76 |
| Bootstrap | 40.04 | 61.94 | 25.69 | 46.65 | 82.11 |
| Truncated | 43.35 | 63.67 | 38.14 | 59.99 | 81.69 |
| Mix-Up | 47.13 | 70.08 | 31.05 | 58.96 | 83.10 |
| Curriculum | 41.81 | 64.53 | 27.57 | 48.84 | 81.68 |
| ELR | 44.95 | 66.65 | 34.66 | 55.72 | 82.62 |
| FCE | 43.81 | 64.97 | 48.85 | 29.92 | 81.82 |
| FCE+B | $\boxed{51.18}$ | $\boxed{73.79}$ | 46.34 | $\boxed{69.92}$ | $\boxed{82.40}$ |
| GCE | 39.81 | 60.51 | 26.93 | 45.17 | 81.13 |
| GCE+B | $\boxed{47.40}$ | $\boxed{71.37}$ | $\boxed{39.13}$ | $\boxed{63.75}$ | 81.37 |
| CE | 39.34 | 61.82 | 25.84 | 46.08 | 81.45 |
| CE+B | 38.47 | $\boxed{61.85}$ | $\boxed{30.00}$ | $\boxed{52.61}$ | 80.72 |
| CEP | 44.39 | 64.56 | 33.33 | 51.45 | 82.06 |
| CEP+B | $\boxed{47.85}$ | $\boxed{71.00}$ | $\boxed{40.56}$ | $\boxed{65.15}$ | 81.79 |

Table 9: Test accuracies obtained by using different losses on the noisy TinyImageNet and Animals10N datasets. Losses implementing the noise-bound are shaded in blue. When using this bound provides benefit, the corresponding value is $\boxed{boxed}$. Overall top values are in **bold**.

### E.3 High and Low Noise Regimes

We conduct additional experiments on the MNIST, FashionMNIST, and CIFAR10 datasets in high ($\eta = 0.8$) and low ($\eta = 0.1$) noise regimes to evaluate our approach in these settings. The results are summarized in Table 10. As with previous tables, we record the clean test accuracy obtained using a loss function (e.g., GCE) and compare this to the clean test accuracy achieved with noise-bounded variants (e.g., GCE+B). Rows corresponding to noise-bounded variants are shaded in blue. We evaluate the SCE, GCE, CE, and FCE loss functions, as well as the L2 loss and its noise-bounded counterpart (L2+B).

In the **high noise regime**, noise-bounding proves highly effective for all loss functions. This is expected, as overfitting is more pronounced at higher noise rates. In the **low noise regime**, we observe minimal advantage in noise-bounding, except for the cross-entropy loss function, which is particularly sensitive to label noise. The GCE, SCE, and FCE loss functions exhibit reasonable robustness to label noise; hence, the impact of noise-bounding is limited at very low noise rates.

| Losses | MNIST | | FashionMNIST | | CIFAR10 | |
|---|---|---|---|---|---|---|
| | 0.1 | 0.8 | 0.1 | 0.8 | 0.1 | 0.8 |
| FCE | $97.54_{\pm 0.73}$ | $90.08_{\pm 0.44}$ | $87.32_{\pm 0.44}$ | $76.91_{\pm 0.55}$ | $89.17_{\pm 0.52}$ | $37.81_{\pm 1.17}$ |
| FCE+B | $97.37_{\pm 0.68}$ | $\mathbf{91.44}_{\pm 0.49}$ | $\mathbf{87.43}_{\pm 1.03}$ | $\mathbf{77.83}_{\pm 0.43}$ | $88.90_{\pm 0.45}$ | $\mathbf{41.56}_{\pm 1.88}$ |
| GCE | $98.44_{\pm 0.08}$ | $78.58_{\pm 0.51}$ | $87.89_{\pm 0.49}$ | $66.17_{\pm 1.48}$ | $89.81_{\pm 0.81}$ | $24.16_{\pm 0.72}$ |
| GCE+B | $98.23_{\pm 0.11}$ | $\mathbf{87.52}_{\pm 0.26}$ | $\mathbf{88.14}_{\pm 0.65}$ | $\mathbf{77.28}_{\pm 0.30}$ | $89.56_{\pm 0.95}$ | $\mathbf{33.05}_{\pm 2.17}$ |
| SCE | $93.34_{\pm 0.53}$ | $42.95_{\pm 2.78}$ | $85.74_{\pm 0.47}$ | $37.05_{\pm 2.76}$ | $87.26_{\pm 0.42}$ | $19.18_{\pm 1.23}$ |
| SCE+B | $\mathbf{97.02}_{\pm 0.35}$ | $\mathbf{44.47}_{\pm 2.58}$ | $85.46_{\pm 0.38}$ | $\mathbf{39.21}_{\pm 1.36}$ | $\mathbf{88.48}_{\pm 0.23}$ | $\mathbf{34.53}_{\pm 1.73}$ |
| CE | $95.86_{\pm 0.40}$ | $45.68_{\pm 1.75}$ | $85.99_{\pm 0.44}$ | $39.97_{\pm 2.09}$ | $83.12_{\pm 1.90}$ | $18.71_{\pm 0.85}$ |
| CE+B | $\mathbf{97.87}_{\pm 0.14}$ | $\mathbf{82.69}_{\pm 1.69}$ | $85.78_{\pm 1.01}$ | $\mathbf{72.70}_{\pm 1.42}$ | $\mathbf{85.77}_{\pm 1.84}$ | $\mathbf{32.60}_{\pm 2.77}$ |
| L2 | $96.81_{\pm 0.17}$ | $44.44_{\pm 0.89}$ | $86.09_{\pm 0.60}$ | $39.28_{\pm 1.07}$ | $88.37_{\pm 1.17}$ | $15.67_{\pm 0.41}$ |
| L2+B | $\mathbf{98.30}_{\pm 0.19}$ | $\mathbf{94.25}_{\pm 0.68}$ | $\mathbf{87.82}_{\pm 0.41}$ | $\mathbf{74.71}_{\pm 0.49}$ | $\mathbf{88.90}_{\pm 0.34}$ | $\mathbf{28.87}_{\pm 1.45}$ |

Table 10: Test accuracies obtained using different losses on the noisy MNIST/ FashionMNIST/ CIFAR10 datasets at high ($\eta = 0.8$) and low noise rates ($\eta = 0.1$). Losses implementing the noise-bound shaded in blue. When using this bound provides benefit, the corresponding value is **bold**. At small noise rates there is minimal gain from noise-bounding whereas at higher noise rate noise-bounding provides significant benefit.

