# OpenReview forum: "Label Noise: Correcting the Forward Correction"
_TMLR — Rejected by TMLR_

### Review · Reviewer_2wLM · 2024-10-18

**Summary Of Contributions:**

This paper studies the forward correction method for noise-label learning. It first proposes a Generalised Forward-Correction that partially unifies forward correction losses and robust loss functions, shedding light on the implicit incorporation of noise model in the later. Further, it proves a lower bound in the Generalised Forward-Correction that is dependent on both noise rate and type of label noise, and proposes a Noise-Bounded Loss for robust learning under a noisy label setting. The actual bound can be computed analytically for different base loss functions. Empirically, when the bound is chosen appropriately under right assumption of label noise, Noise-Bounded Loss is beneficial in improving neural network performance under a noise-label learning setting.

**Audience:**

Yes

**Broader Impact Concerns:**

N.A.

**Claims And Evidence:**

Yes

**Requested Changes:**

Comments for authors' consideration:

C1. The noise bound is dependent on both noise rate and type of label noise. However, the ablation study has only demonstrated the impact of assumed noise rate on model performance through the computed noise bound. It would be good to incorporate an additional ablation study. In this experiment, the noise bound should be ablated on different types of label noise while being conditioned on a fixed noise rate. It could facilitate the understanding of the impact of implemented noise bound on model performance under different label noise assumptions.

C2. In Figure.2: It would be good to incorporate a horizontal line indicating the validation accuracy of the baseline model trained with the base loss only. It would clearly show the effective range of noise rate that leads to improved validation accuracy than the baseline model.

Trivial - Ignore if irrelevant:

In Equation.~1 - FCE: $\hat{T}$ is applied with a parenthesis here, implying it being a function. However, it is represented as a stochastic matrix throughout the paper. It would be better to standardise its usage.

Potential typo in Paragraph "Forward-Corrections" - 6$^\text{th}$ line: "high" $\rightarrow$ "low". As current description using "high" seems to contradict the second last sentence in the same paragraph and the values in the table below.

**Strengths And Weaknesses:**

Strengths:

S1. This paper proposes a more general framework for forward-correction and partially include robust losses in noise-label learning. It reveals the implicit incorporation of a noise model as the fundamental mechanism to alleviate overfitting to label noises.

S2. This paper provides a Noise-Bound Loss as a solution to achieve robust network training under the noise label setting, where the bound can be analytically computed for different base loss functions with prior knowledge of noise rate and label noise type.

S3. Empirical results demonstrate that Noise-Bound Loss, if the bound is carefully selected, is beneficial to model performance in most of the cases when applied with different base losses.

S4. The whole paper is logically presented and well written, with examples showing connections between the proposed general framework and existing methods.

S5. The notations are clear and followed throughout the paper without much ambiguity.

Weakness:

W1. Effectiveness of the proposed Noise-Bound Loss is dependent on the prior knowledge (or accurate assumption) on both rate and type of label noise, which are mostly unknown.

---

> ### Author Response · Authors · 2024-11-02
> **Reponse to Reviewer 2wLM**
>
> Dear Reviewer,
>
> We greatly appreciate the time you have taken to review our paper. We value your comments and will make the relevant alterations.
>
> **C1:** We agree that an ablation study focusing on different types of label noise, while keeping the noise rate fixed, would improve our understanding of the noise-bound's impact under varying conditions.
>
> **C2:** Incorporating a horizontal line in Figure 2 to indicate the validation accuracy of the baseline model makes a great deal of sense and would be a worthy inclusion in our opinion.
>
> **Typos:** Thank you for spotting these errors. We will correct them in the next version of our manuscript.
>
> **Proposed Changes:**
>
> 1) **Ablation Study Addition:** We will include an ablation study that evaluates the effectiveness of our approach when the noise bound is computed for different (non-symmetric) noise models, with the noise rate remaining constant.
>
> 2) **Figure Enhancement:** A horizontal line will be added to Figure 2 to provide a baseline comparison and better contextualise the results.
>
> Thank you again for your constructive feedback.

---

### Review · Reviewer_DMsM · 2024-10-23

**Summary Of Contributions:**

(1) This paper addresses the challenge of learning with label noise by introducing a "noise-bound" on top of the forward-correction loss function.

(2) The authors provide both theoretical and empirical evidence demonstrating the effectiveness of this noise-bound in preventing overfitting.

(3) The paper also generalizes the forward-correction method to include non-linear models, broadening its applicability.

**Audience:**

Yes

**Claims And Evidence:**

Yes

**Requested Changes:**

1. In Table 3 and 4, please include the results for high noise rate (e.g. 0.6 and 0.8).

2. Could the authors provide further discussion on scenarios where the proposed noise-bound might be less effective? While the ineffectiveness in the asymmetric case is somehow expected, an analysis of when and why the noise-bound underperforms would be beneficial for understanding its limitations and potential applications.

**Strengths And Weaknesses:**

(S1) The paper is well-written and easy to follow, with the background section offering clear and informative context.

(S2) The proposed method is well-motivated. The idea of lower bounding the forward-correction loss is intuitive and backed by theoretical analysis under given certain conditions.

(S3) The method is simple and efficient to implement, making it a practical plug-and-play module for a variety of applications.


(W1) The derivation of the proposed lower bound relies on the assumption of symmetric label noise, which limits its applicability in more complex real-world scenarios involving asymmetric noise.

(W2) Table 3/4 shows that applying the noise-bound does not lead to consistent performance improvements, even in symmetric noise conditions. Additionally, results for settings with high noise rates are absent.

(W3) The performance gains from the proposed method appear to be marginal, with many results falling within the standard deviation, making it unclear how much practical benefit is gained.

---

> ### Author Response · Authors · 2024-11-02
> **Reponse to Reviewer DMsM**
>
> Dear Reviewer,
> Thank you for your thoughtful review. We value your input and have carefully considered your comments.
>
> You have requested further discussion on scenarios where the proposed noise-bound might be less effective. While we have touched upon this topic in Section D.2 of our paper, we acknowledge that it could be clearer and more detailed as you suggested. We are committed to expanding this discussion to better illustrate the limitations and potential applications of our approach.
>
> **Proposed Changes:**
>
> 1) **Higher Noise Rates:** We will include results for higher noise rates, e.g., 0.6 and 0.8..
>
> 2) **Expanded Discussion:** We will enhance the discussion on scenarios where the noise-bound may underperform.

---

### Review · Reviewer_L8Gc · 2024-10-26

**Summary Of Contributions:**

This paper proposed an algorithm to tackle overfitting when the dataset contains noise labels. Specifically, the authors proposed the noise-bounded loss, an improvement from the forward-correction method, for training the machine learning model (neural network in the experiments) to avoid overfitting due to noise labels.

**Audience:**

Yes

**Claims And Evidence:**

No

**Requested Changes:**

For point 1, the result for the test data comes from the same distribution as training data is expected, e.g., the test data should not only contain clean data but also noise data too.

For point 2, additional traditional overfitting-preventing method is expected to evaluate the effectiveness of the proposed method.

For point 4, the result of a lower noise rate is expected since the noise rate will be "closer" to the real-world scenarios.

Also, any result/insight for points 3 and 5 is expected to increase the significance of the contribution.

**Strengths And Weaknesses:**

Strengths:
1. The noise label data exists in almost all real-world datasets, and it is essential to design an algorithm to solve the overfitting issue when the noise label.
2. The paper provided a theoretical analysis of the "Risk Bounds" under two assumptions.
3. The experiments show the improvement of the proposed method from the forward-correction method under the assumption of known approximate label noise rate and the current setting of the experiment.

Weaknesses:
1.  The current experiment's setting is training the model on the noise data and testing on the clean data. I feel this is very confusing, and I think keeping the testing and training data from the same distribution is very important. It will be unrealistic to know whether the coming unknown data is clean. The testing data from the same distribution as the training data should be the setting for all the experiments.
2. None of the traditional methods to avoid overfitting is provided within the experiments and the comparison settings. For example, the dropout, cross-validation, early stopping, or even splitting the dataset into the train-validation-test could prevent overfitting when selecting the best model on the validation set. It is very important to compare this with the traditional overfitting-preventing method to fully evaluate this paper's contribution.
3. The bounded loss, which is the major contribution of this paper, requires the (approximate) label noise rate to be known. This is an optimistic assumption for the practical cases. It requires prior knowledge of which data's label is noised to estimate the label noise rate. However, one can simply remove the noise data when the prior knowledge is known, which does not require additional methods to avoid overfitting issues from noise labels.
4. The result compared to other loss functions shows that the proposed method does not provide the best result for half of the comparisons that the author provided (Total comparison on 7 datasets are provided, and experiments show 3.5 of them show the  "+B" works the best). Also, the assumed label noise rates (0.2, 0.4, and 0.6) are high, which is impractical in real-world settings. To provide a convincing result, smaller noise rates, which are expected to be closer to the practical settings, are expected.
5. (Please correct me if I am wrong on this point) Following the point 4 above, the forward-correction and/or with the bounded noise seems to only work for cross-entropy loss in general. While cross-entropy is important in many machine-learning settings, it limited the paper's contribution from my perspective. In other words, the contribution would be significantly increased if the author provided the bounded noise to other losses as well.

---

> ### Author Response · Authors · 2024-11-02
> **Reviewer L8Gc Reponse**
>
> Dear reviewer, thank you kindly for taking the time to review our paper. We appreciate the effort although we would like to address a few major misconceptions in your comments.
>
> 1) **Evaluating Noise-Robust Approaches:** Evaluating the effectiveness of a noise-robust approach requires testing with cleanly labelled data. This allows us to determine if the approach genuinely improves model performance, a standard methodology cited across numerous studies, such as [1], [2], [3]. We believe this critique may overlook the necessity of such an evaluation method, which is a common practice in the field.
>
>     - [1] Co-teaching: Robust training of deep neural networks with extremely noisy labels (2018) Han et al
>     - [2] MentorNet: Learning Data-Driven Curriculum for Very Deep Neural Networks on Corrupted Labels (2018) Jiang et al
>     - [3] DivideMix: Learning with Noisy Labels as Semi-supervised Learning (2020) Hoi et al
>
> 2) **Baselines:** Our paper is designed to show that noise-bounding a loss will result in improved robustness. Our work is largely orthogonal to previous work such as [1],[2],[3] and may be used in conjunction with these approaches which is why this comparison is absent. We can include other approaches in the appendix for context if you strongly believe this would provide a useful contextualisation of our work.
>
> 3) **Misunderstanding of Noise Rate Knowledge:** Our method requires a rough idea of the noise rate; for example it would be sufficient to know that '_roughly 20% of the labels are noisy_'. It does not require knowledge of which data instances are noisy: This constitutes a fundamental misunderstanding on your part. Knowledge of the approximate noise _rate_ is not sufficient to denoise the data as you claim.
>
> 4) **Effectiveness of the '+B' Approach:** Our findings do not support the notion that the ‘+B’ approach works in only 3.5 out of 7 datasets. Our data shows that noise-bounding is beneficial in the majority of scenarios. The chosen noise rates of 0.2, 0.4, and 0.6, while high, align with those commonly tested in the field to demonstrate robustness in the high noise regime.
>
> 5) **Applicability to Other Loss Functions:** Although our approach is effective with the cross-entropy loss, it also applies to symmetric and generalized cross-entropy losses, which differ significantly from standard cross-entropy despite the similar name. However in light of you comment we are willing to include the L2 loss function if you think this would provide a useful contribution. Please let us know your thoughts.
>
> **Proposed Changes:**
>
> - **For point 2:** We will include comparisons with traditional methods for preventing overfitting to better contextualise our results.
> - **For point 3:** We plan to conduct additional experiments with lower noise rates (e.g., 0.1) to reflect more typical real-world scenarios.
> - **For point 5:** We will explore the implementation of a forward-corrected L2 loss function incorporating our proposed noise-bound, as suggested.

---

### Decision · Action_Editor_w4ma · 2024-12-09

**Recommendation:** Reject

**Comment:**

After considering the reviewers' feedback and the authors' responses, the recommendation is to reject this submission. There are following unresolved issues raised during the author-reviewer discussion:

- Key Comparisons Missing: Comparisons to standard overfitting prevention techniques, such as early stopping or dropout, are absent, as noted by Reviewer L8Gc. Furthermore, results for high-noise regimes requested by Reviewer DMsM remain unaddressed.

- Practical Limitations: Both Reviewer 2wLM and Reviewer DMsM pointed out the dependency on prior knowledge of noise rates, which is often unavailable in real-world scenarios, limiting the method’s utility.

- Inconsistent Results: As highlighted by Reviewer DMsM, the proposed noise-bound method does not consistently outperform existing approaches and its reported performance gains are often within the standard deviation.



- Insufficient Analysis of Failure Cases: Reviewer DMsM remarked that "there is insufficient discussion of the failure cases for the proposed bound, which undermines its practical reliability." A thorough analysis of the limitations and scenarios where the method underperforms remains absent.

While the submission targets an important problem in machine learning, its contributions and the empirical evaluation is insufficiently robust to support its claims and attract audiences. Addressing the issues outlined above in future revisions, such as including comprehensive baseline comparisons, exploring practical scenarios with realistic noise assumptions, and presenting a deeper analysis of failure cases, could significantly strengthen the paper.

**Audience:**

While the topic of avoiding overfitting in noisy-label learning is generally of interest to TMLR's audience, the paper's fail to sufficiently stand out due to the following:

- Practical Relevance: Reviewer L8Gc raised concerns about the practicality of the method, stating, "The bounded loss requires the (approximate) label noise rate to be known. This is an optimistic assumption for practical cases." This reliance on a difficult-to-estimate parameter reduces the method’s real-world applicability.

- Inconsistent Results: As Reviewer DMsM observed, "Table 3/4 shows that applying the noise-bound does not lead to consistent performance improvements, even in symmetric noise conditions." This inconsistency weakens the claim that the method reliably improves robustness.

- Incremental Contribution: Reviewer DMsM remarked, "The method proposed by this paper is largely incremental." We think that adding a lower bound to prevent loss minimization is conceptually straightforward and does not represent a substantial advance over existing methods like early stopping or validation-based selection. This limits its ability to attract interest from the audience.

**Claims And Evidence:**

The claims made in this submission are not consistently supported by accurate, convincing, and clear evidence. Specifically:

- Dependence on Noise Assumptions: Reviewer 2wLM highlighted, "Effectiveness of the proposed Noise-Bound Loss is dependent on the prior knowledge on both rate and type of label noise, which are mostly unknown." While the authors responded that "it would be sufficient to know that roughly 20% of the labels are noisy" and stated that it does not require knowledge of which specific instances are noisy, we believe it is generally unrealistic to estimate such an average without examining the data. This reliance on prior knowledge substantially limits the practicality of the proposed method.

- Incomplete Baseline Comparisons: Reviewer L8Gc noted, "None of the traditional methods to avoid overfitting is provided within the experiments and the comparison settings." The absence of comparisons with standard techniques such as dropout, cross-validation, or early stopping weakens the empirical evaluation and raises questions about the novelty and relevance of the proposed method.